# Meta-Learning Stationary Stochastic Process Prediction with Convolutional Neural Processes

**Andrew Y. K. Foong**[*]
University of Cambridge
ykf21@cam.ac.uk

**Wessel P. Bruinsma**[*]
University of Cambridge
Invenia Labs
wpb23@cam.ac.uk

**Jonathan Gordon**[*]
University of Cambridge
jg801@cam.ac.uk

**Yann Dubois**
University of Cambridge
yanndubois96@gmail.com

**James Requeima**
University of Cambridge
Invenia Labs
jrr41@cam.ac.uk

**Richard E. Turner**
University of Cambridge
Microsoft Research
ret26@cam.ac.uk

## Abstract

Stationary stochastic processes (SPs) are a key component of many probabilistic models, such as those for off-the-grid spatio-temporal data. They enable the statistical symmetry of underlying physical phenomena to be leveraged, thereby aiding generalization. Prediction in such models can be viewed as a *translation equivariant* map from observed data sets to predictive SPs, emphasizing the intimate relationship between stationarity and equivariance. Building on this, we propose the Convolutional Neural Process (ConvNP), which endows Neural Processes (NPs) with translation equivariance and extends convolutional conditional NPs to allow for dependencies in the predictive distribution. The latter enables ConvNPs to be deployed in settings which require coherent samples, such as Thompson sampling or conditional image completion. Moreover, we propose a new maximum-likelihood objective to replace the standard ELBO objective in NPs, which conceptually simplifies the framework and empirically improves performance. We demonstrate the strong performance and generalization capabilities of ConvNPs on 1D regression, image completion, and various tasks with real-world spatio-temporal data.

## 1 Introduction

Incorporating appropriate inductive biases into machine learning models is key to achieving good generalization performance. Consider, for example, predicting rainfall at an unseen test location from rainfall measurements nearby. A powerful inductive bias for this task is *stationarity*: the assumption that the generative process governing rainfall is spatially homogeneous. Given only observations in a limited part of the space, stationarity allows the model to extrapolate to yet unobserved regions. Closely related to stationarity is *translation equivariance* (TE). TE formalizes the intuitive idea that if observations are shifted in time or space, then the resulting predictions should be shifted by the same amount. When stationarity or TE is appropriate, e.g. in time-series [31], images [22], and spatio-temporal modelling [8, 7], incorporating them into our models yields significant benefits.

A general framework for these tasks is to view them as prediction of a *stochastic process* (SP; [32]). This principled approach has inspired a new set of deep learning architectures that bring the expressivity and fast test-time inference of deep learning to SP modelling. *Conditional Neural Processes* (CNPs; [10]) use neural networks to directly parameterize a map from data sets to predictive

---

[*]Authors contributed equally.

SPs, which is trained via meta-learning [34, 39]. However, CNPs suffer from several drawbacks that inhibit their use in scenarios where other SP models, e.g. Gaussian processes (GPs; [29]), often succeed. First, vanilla CNPs cannot account for TE as an inductive bias. This was recently addressed with the introduction of ConvCNPs [13]. Second, both CNPs and ConvCNPs are limited to factorized, parametric predictive distributions. This makes them unsuitable for producing coherent predictive function samples or modelling complicated likelihoods. *Neural Processes* (NPs; [11]), a latent variable extension of CNPs, were introduced to enable richer joint predictive distributions. However, the NP training procedure uses variational inference (VI) and amortization, which are known to suffer from certain drawbacks [40, 6]. Moreover, existing NPs do not incorporate TE.

This paper builds on ConvCNPs and NPs [11, 13] to develop *Convolutional Neural Processes* (ConvNPs). ConvNPs are a map from data sets to predictive SPs that is both TE *and* capable of expressing complex joint distributions. As training ConvNPs with VI poses technical and practical issues, we instead propose a simplified maximum-likelihood objective, which directly targets the predictive SP. We show that ConvNPs produce compelling samples and generalize effectively, making them suitable for a broad range of spatio-temporal prediction tasks. Our key contributions are:

1. We introduce ConvNPs, extending ConvCNPs to model rich joint predictive distributions.
2. We propose a simplified training procedure, discarding VI in favor of an approximate maximum-likelihood procedure, which improves performance for ConvNPs.
3. We demonstrate the usefulness of ConvNPs on toy time-series experiments, image-based sampling and extrapolation, and real-world environmental data sets.

## 2 Problem Set-up and Background

**Notation.** The main paper provides an informal treatment of ConvNPs. We refer the reader to the supplement for precise definitions and statements. Let $\mathcal{X} = \mathbb{R}^{d_{\mathrm{in}}}, \mathcal{Y} = \mathbb{R}$ denote the input and output spaces, and let $(\boldsymbol{x}, y)$ be an input-output pair. Let $\mathcal{S}$ be the collection of all finite data sets, with $D_c, D_t \in \mathcal{S}$ a *context* and *target* set respectively. We will later consider predicting the target set from the context set as in [10, 11]. Let $\boldsymbol{X}_c, \boldsymbol{y}_c$ be the inputs and corresponding outputs of $D_c$, with $\boldsymbol{X}_t, \boldsymbol{y}_t$ defined analogously. We denote a single *task* as $\xi = (D_c, D_t) = ((\boldsymbol{X}_c, \boldsymbol{y}_c), (\boldsymbol{X}_t, \boldsymbol{y}_t))$. Let $\mathcal{P}(\mathcal{X})$ denote the collection of stochastic processes on $\mathcal{X}$, and let $C_b(\mathcal{X})$ denote the collection of continuous, bounded functions on $\mathcal{X}$.

### 2.1 Meta-Learning Stochastic Process Prediction

Consider rainfall $y$ as a function of position $\boldsymbol{x}$. To model rainfall, we can view it as a *random* function from $\mathcal{X}$ to $\mathcal{Y}$. Mathematically, this corresponds to a SP on $\mathcal{X}$—a probability distribution over functions from $\mathcal{X}$ to $\mathcal{Y}$—which we denote by $P$. Given perfect knowledge of $P$, we could predict rainfall at any location of interest by conditioning $P$ on observations $D_c$, yielding a *predictive* SP. However, in practice we will only have access to a large collection of sample functions from $P$. Each function is known only at a finite set of inputs, $D = (\boldsymbol{x}_n, y_n)_{n=1}^{N}$, which we divide into $D_c, D_t$ for meta-training. Given sufficient data, we can *meta-learn* the map from context sets $D_c$ to the ground-truth predictive distribution: $D_c \mapsto p(\boldsymbol{y}_t | \boldsymbol{X}_t, D_c) = p(\boldsymbol{y}_t, \boldsymbol{y}_c | \boldsymbol{X}_t, \boldsymbol{X}_c)/p(\boldsymbol{y}_c | \boldsymbol{X}_c)$. As long as the predictives for varying $\boldsymbol{X}_t$ are Kolmogorov-consistent [37, Section 2.4], this corresponds to learning a map from data sets directly to *predictive* SPs. We refer to the map that takes a context set $D_c$ to the *exact* ground truth SP conditioned on $D_c$ as the *prediction map* $\pi_P \colon \mathcal{S} \to \mathcal{P}(\mathcal{X})$ (details in App A). The general prediction problem may then be viewed as learning to approximate $\pi_P$.

### 2.2 Translation Equivariance and Stationarity

The prediction map $\pi_P$ possesses two important symmetries. First, $\pi_P$ is *invariant* to permutations of $D_c$ [43, 13]. Second, if the ground truth process $P$ is *stationary*, then $\pi_P$ is *translation equivariant*: whenever an input to the map is translated, its output is translated by the same amount (see App B for formal definitions and proofs). This simple statement highlights the intimate relationship between stationarity and TE. Moreover, it suggests that models for the prediction map should also be TE and permutation invariant. As such models are a small subset of the space of *all* models, building in these properties can greatly improve data efficiency and generalization for stationary SP prediction. In Sec 3, we extend the TE maps of Gordon et al. [13] (reviewed next) to construct a rich class of models which incorporate these inductive biases.

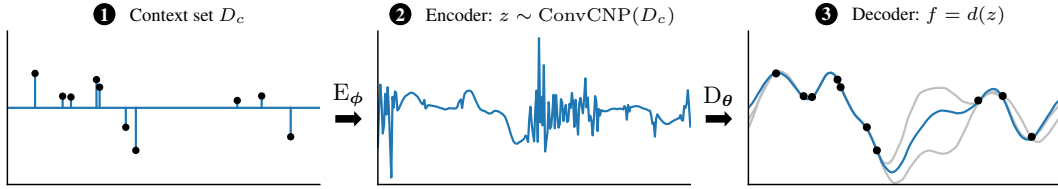

Figure 1: ConvNP encoder-decoder architecture. The encoder is a ConvCNP which takes the context set as input (left panel) and outputs a single sample of $z$ (center panel). The decoder takes this as input and outputs a predictive sample (right panel blue; two other samples shown in grey).

## 2.3 Convolutional Conditional Neural Processes

We review ConvCNPs [13], which are an important building block in our proposed model. ConvCNPs can be viewed from the perspective of SP prediction, revealing their key limitations. Given a context set $D_c$, the ConvCNP models the predictive distribution over target outputs as:

$$p_{\boldsymbol{\phi}}(\boldsymbol{y}_t | \boldsymbol{X}_t, D_c) = \prod_{(\boldsymbol{x}, y) \in D_t} \mathcal{N}(y; \mu(\boldsymbol{x}, D_c), \sigma^2(\boldsymbol{x}, D_c)). \tag{1}$$

The mean $\mu(\,\cdot\,, D_c)$ and variance $\sigma^2(\,\cdot\,, D_c)$ are parametrized by *convolutional deep sets* (ConvDeepSets; [13]): a flexible parametrization for TE maps from $\mathcal{S}$ to $C_b(\mathcal{X})$. ConvDeepSets introduce the idea of *functional representations*: whereas the standard DeepSets framework embeds data sets into a finite-dimensional vector space [43], a ConvDeepSet embeds data sets in an infinite-dimensional function space. ConvDeepSets are a composition of two stages. The first stage maps a data set $D$ to its functional representation via $D \mapsto \sum_{(\boldsymbol{x}, y) \in D} \phi(y) \psi(\,\cdot\, - \boldsymbol{x})$. Here $\phi(y) = (1, y) \in \mathbb{R}^2$ and $\psi$ is the Gaussian radial basis function. This functional representation is then passed to the second stage, a TE map between function spaces, implemented by a convolutional neural network (CNN). See App C for a full description of the ConvCNP including pseudocode.

We observe that Eq (1) defines a map from context sets $D_c$ to predictive SPs. Specifically, let $\mathcal{P}_N(\mathcal{X}) \subset \mathcal{P}(\mathcal{X})$ denote the set of *noise GPs*: GPs on $\mathcal{X}$ whose covariance is given by $\mathrm{Cov}(\boldsymbol{x}, \boldsymbol{x}') = \sigma^2(\boldsymbol{x})\delta[\boldsymbol{x} - \boldsymbol{x}']$, where $\sigma^2 \in C_b(\mathcal{X})$ and $\delta[0] = 1$ with $\delta[\,\cdot\,] = 0$ otherwise. Then the ConvCNP is a map $\mathrm{ConvCNP} : \mathcal{S} \to \mathcal{P}_N(\mathcal{X})$ with Eq (1) defining its finite-dimensional distributions. Since ConvDeepSets are TE, and the means and variances of ConvCNPs are ConvDeepSets, it follows that ConvCNPs are also TE as maps from $\mathcal{S} \to \mathcal{P}_N(\mathcal{X})$ (see App D for a more formal derivation). Unfortunately, processes in $\mathcal{P}_N(\mathcal{X})$ possess two key limitations. First, it is impossible to obtain coherent function samples as each point of the function is generated independently. Second, Gaussian distributions cannot model multi-modality, heavy-tailedness, or asymmetry.

## 3 The Convolutional Neural Process

We now present the ConvNP, which addresses the weaknesses of ConvCNPs. We introduce their parametrization (Sec 3.1) and a maximum-likelihood meta-training procedure (Sec 3.2).

### 3.1 Parametrizing Translation Equivariant Maps to Stochastic Processes Using ConvNPs

The ConvNP extends the ConvCNP by parametrizing a map to predictive SPs more expressive than $\mathcal{P}_N(\mathcal{X})$, allowing for coherent sampling and non-Gaussian predictives. It achieves this by passing the output of a ConvCNP through a non-linear, TE map between function spaces. Specifically, the ConvNP uses an encoder–decoder architecture, where the encoder $\mathrm{E} : \mathcal{S} \to \mathcal{P}_N(\mathcal{X})$ is a ConvCNP and the decoder $d : \mathbb{R}^{\mathcal{X}} \to \mathbb{R}^{\mathcal{X}}$ is TE (here $\mathbb{R}^{\mathcal{X}}$ denotes the set of all functions from $\mathcal{X}$ to $\mathbb{R}$). Conditioned on $D_c$, ConvNP samples can be obtained by sampling a function $z \sim \mathrm{ConvCNP}(D_c)$ and then computing $f = d(z)$. This is illustrated in Fig 1. Importantly, $d$ takes functions to functions and does not necessarily act point-wise: letting $f(\boldsymbol{x})$ depend on the value of $z$ at multiple locations is crucial for inducing dependencies in the predictive. This sampling procedure induces a map between SPs, $\mathrm{D} : \mathcal{P}_N(\mathcal{X}) \to \mathcal{P}(\mathcal{X})$ (see App D). Putting these together, with explicit parameter dependence in E and D, the ConvNP is constructed as

$$\mathrm{ConvNP}_{\boldsymbol{\theta}, \boldsymbol{\phi}} = \mathrm{D}_{\boldsymbol{\theta}} \circ \mathrm{E}_{\boldsymbol{\phi}}, \quad \mathrm{E}_{\boldsymbol{\phi}} = \mathrm{ConvCNP}_{\boldsymbol{\phi}}, \quad \mathrm{D}_{\boldsymbol{\theta}} = (d_{\boldsymbol{\theta}})_*,$$

where $(d_{\boldsymbol{\theta}})_*$ is the pushforward[2] under $d_{\boldsymbol{\theta}}$. In App D, we prove that $\mathrm{ConvNP}_{\boldsymbol{\theta}, \boldsymbol{\phi}}$ is indeed TE.

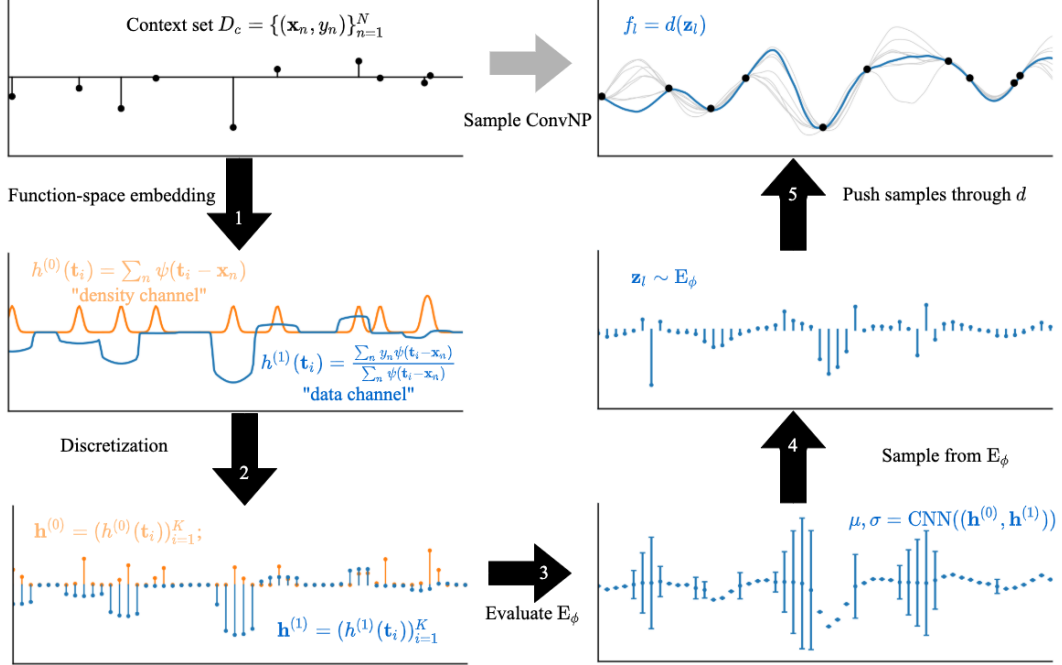

Figure 2: Forward pass of a ConvNP. Steps (1)-(4) depict sampling from the encoder $E_\phi$, which is a ConvCNP. This involves: (1) computing a functional representation of the context set, with separate 'density' and 'data' channels (described in detail in Gordon et al. [13] and App C), (2) discretizing the representation, (3) passing the representation through a CNN, which outputs the parameters of independent Gaussian distributions spaced on a grid, and (4) sampling from these distributions. However, the samples at each grid point are independent of each other, hence in (5) the samples are passed through *another* CNN, the decoder, to induce dependencies, and then are smoothed out.

In practice, we cannot compute samples of noise GPs ($\mathcal{P}_N$) because they comprise uncountably many independent random variables. Instead, we consider a discrete version of the model, which enables computation. Following Gordon et al. [13], we discretize the domain of $z$ on a grid $(\boldsymbol{x}_i)_{i=1}^K$, with $\boldsymbol{z} := (z(\boldsymbol{x}_i))_{i=1}^K$. As a consequence, the model can only be equivariant up to shifts on this discrete grid. With this discretization, sampling $\boldsymbol{z} \sim \text{ConvCNP}_{\boldsymbol{\phi}}(D_c)$ amounts to sampling independent Gaussian random variables, and $d_{\boldsymbol{\theta}}$ is implemented by passing $\boldsymbol{z}$ through a CNN. The forward pass of a trained ConvNP is illustrated in Fig 2. Note that CNNs are not always entirely TE due to the zero padding that occurs at each layer. In practice, we find that this is not an issue.[3] Following Kim et al. [14], we define the model likelihood by adding heteroskedastic Gaussian observation noise $\sigma_y^2(\boldsymbol{x}, \boldsymbol{z})$ to the predictive function draws $f = d_{\boldsymbol{\theta}}(\boldsymbol{z}) \in \mathbb{R}^{\mathcal{X}}$:

$$p_{\boldsymbol{\phi}, \boldsymbol{\theta}}(\boldsymbol{y}_t | \boldsymbol{X}_t, D_c) = \mathbb{E}_{\boldsymbol{z} \sim E_{\boldsymbol{\phi}}(D_c)} \left[ \prod_{(\boldsymbol{x}, y) \in D_t} \mathcal{N}\left(y; d_{\boldsymbol{\theta}}(\boldsymbol{z})(\boldsymbol{x}), \sigma_y^2(\boldsymbol{x}, \boldsymbol{z})\right) \right]. \tag{2}$$

Although the product in the expectation factorizes, $p_{\boldsymbol{\phi}, \boldsymbol{\theta}}(\boldsymbol{y}_t | \boldsymbol{X}_t, D_c)$ does not: $\boldsymbol{z}$ induces dependencies in the predictive, in contrast to Eq (1). See App C for full implementation details for the ConvNP.

### 3.2 Maximum Likelihood Learning of ConvNPs

We now propose a maximum-likelihood training procedure for ConvNPs. Let the ground truth task distribution be $p(\xi) = p(D_c, D_t)$. Let $\mathcal{L}_{\text{ML}}(\boldsymbol{\theta}, \boldsymbol{\phi}; \xi) := \log p_{\boldsymbol{\phi}, \boldsymbol{\theta}}(\boldsymbol{y}_t | \boldsymbol{X}_t, D_c)$ be the single-task likelihood, and let $\mathcal{L}_{\text{ML}}(\boldsymbol{\theta}, \boldsymbol{\phi}) := \mathbb{E}_{p(\xi)}[\log p_{\boldsymbol{\phi}, \boldsymbol{\theta}}(\boldsymbol{y}_t | \boldsymbol{X}_t, D_c)]$ be the task-averaged likelihood. The following proposition shows that maximizing $\mathcal{L}_{\text{ML}}$ recovers the prediction map $\pi_P$ in a suitable limit:

**Prop 1.** Let $\Psi: \mathcal{S} \rightarrow \mathcal{P}(\mathcal{X})$ be a map from data sets to SPs, and let $\mathcal{L}_{\text{ML}}(\Psi) := \mathbb{E}_{p(\xi)}[\log p_\Psi(\boldsymbol{y}_t | \boldsymbol{X}_t, D_c)]$ where $p_\Psi$ is the density of $\Psi(D_c)$ at $\boldsymbol{X}_t$. Then $\Psi$ globally maximizes $\mathcal{L}_{\text{ML}}(\Psi)$ if and only if $\Psi = \pi_P$. See App E for more details and conditions.

In practice, we do not have infinite flexibility in our model or infinite data to compute expectations over $p(\xi)$, but Prop 1 shows that maximum-likelihood training is sensible with an expressive model and sufficient data. Letting $\mathcal{D} = \{\xi_n\}_{n=1}^{N_{\text{tasks}}}$ be a *meta-training* set, we can train a ConvNP by stochastic gradient maximization of $\mathcal{L}_{\text{ML}}$ with tasks sampled from $\mathcal{D}$. Unfortunately, for non-linear decoders, $\log p_{\phi,\theta}(\boldsymbol{y}_t|\boldsymbol{X}_t, D_c)$ is intractable due to the expectation over $\boldsymbol{z}$ (Eq (2)). For a given task $\xi$, we instead optimize the following Monte Carlo estimate of $\mathcal{L}_{\text{ML}}(\boldsymbol{\theta}, \boldsymbol{\phi}; \xi)$, which is conservatively biased, consistent, and monotonically increasing in $L$ (in expectation) [3]:

$$\hat{\mathcal{L}}_{\text{ML}}(\boldsymbol{\theta}, \boldsymbol{\phi}; \xi) \coloneqq \log \left[ \tfrac{1}{L} \sum_{l=1}^{L} \exp \left( \sum_{(\boldsymbol{x},y) \in D_t} \log p_{\boldsymbol{\theta}}(y|\boldsymbol{x}, \boldsymbol{z}_l) \right) \right]; \quad \boldsymbol{z}_l \sim \text{E}_{\boldsymbol{\phi}}(D_c). \tag{3}$$

One drawback of this objective is that single sample estimators are not useful, as they drive $\boldsymbol{z}$ to be deterministic. In our experiments, we set $L$ between 16 and 32. For further discussion of the effect of $L$ see App G. Eq (3) can be viewed as importance sampling in which the prior is the proposal distribution. Prior sampling is typically ineffective as it is unlikely to propose functions that pass near observed data. Here, however, $\text{E}_{\boldsymbol{\phi}}$ depends on context sets $D_c$, which often is sufficient to constrain prior function samples to be close to $D_t$. In Sec 5, we demonstrate that, perhaps surprisingly, this estimator often significantly outperforms VI-inspired estimators (discussed next).

# 4 The Latent Variable Interpretation of ConvNPs

We now describe an alternative approach to training the ConvNP via variational lower bound maximization. This serves the dual purpose of relating ConvNPs to the NP family, and contrasting the existing NP framework with our simplified, maximum-likelihood approach from Sec 3.2.

## 4.1 A Variational Lower Bound Approach to ConvNPs

Garnelo et al. [11] propose viewing Neural Processes as performing approximate Bayesian inference and learning in the following latent variable model:

$$\boldsymbol{z} \sim p_{\boldsymbol{\theta}}(\boldsymbol{z}); \quad y(\boldsymbol{x}) = f_{\boldsymbol{\theta}}(\boldsymbol{x}; \boldsymbol{z}); \quad p_{\boldsymbol{\theta}}(\boldsymbol{y}_t|\boldsymbol{X}_t, \boldsymbol{z}) = \prod_{(\boldsymbol{x},y) \in D_t} \mathcal{N}\left(y; f_{\boldsymbol{\theta}}(\boldsymbol{x}; \boldsymbol{z}), \sigma_y^2\right). \tag{4}$$

To train the model, they propose using *amortized* VI [15, 30]. This approach involves introducing a variational approximation $q_{\boldsymbol{\phi}}$ which maps data sets $S \in \mathcal{S}$ to distributions over $\boldsymbol{z}$, and maximizing a lower bound (ELBO) on $\log p_{\boldsymbol{\theta}}(\boldsymbol{y}_t|\boldsymbol{X}_t, D_c)$. We can define a similar procedure for ConvNPs. For ConvNPs, $\boldsymbol{z}$ is a latent *function*, $q_{\boldsymbol{\phi}}$ is a map from data sets to SPs, and $f_{\boldsymbol{\theta}}$ is a map between function spaces. A natural choice is to use a ConvCNP and CNN for $q_{\boldsymbol{\phi}}$ and $f_{\boldsymbol{\theta}}$, respectively. This results in the same parameterization as in Sec 3, but a different modelling interpretation and meta-training objective. Given a task $\xi = (D_c, D_t)$, the ELBO for this model is:

$$\mathbb{E}_{\boldsymbol{z} \sim q_{\boldsymbol{\phi}}(\boldsymbol{z}|D_c \cup D_t)} \left[\log p_{\boldsymbol{\theta}}(\boldsymbol{y}_t|\boldsymbol{X}_t, \boldsymbol{z})\right] - \text{KL}(q_{\boldsymbol{\phi}}(\boldsymbol{z}|D_c \cup D_t)\|p(\boldsymbol{z}|D_c)).$$

As $p(\boldsymbol{z}|D_c)$ is intractable to compute, Garnelo et al. [11] instead propose the following objective:

$$\mathcal{L}_{\text{NP}}(\boldsymbol{\theta}, \boldsymbol{\phi}; \xi) \coloneqq \mathbb{E}_{\boldsymbol{z} \sim q_{\boldsymbol{\phi}}(\boldsymbol{z}|D_c \cup D_t)} \left[\log p_{\boldsymbol{\theta}}(\boldsymbol{y}_t|\boldsymbol{X}_t, \boldsymbol{z})\right] - \text{KL}(q_{\boldsymbol{\phi}}(\boldsymbol{z}|D_c \cup D_t)\|q_{\boldsymbol{\phi}}(\boldsymbol{z}|D_c)), \tag{5}$$

where the intractable term $p(\boldsymbol{z}|D_c)$ has been substituted with our variational approximation $q_{\boldsymbol{\phi}}(\boldsymbol{z}|D_c)$. Due to this substitution, $\mathcal{L}_{\text{NP}}$ is no longer a valid ELBO for the original model (Eq (4)). Rather, if we define *separate* models for each context set $D_c$, and *define* the conditional prior for each model as $p(\boldsymbol{z}|D_c) \coloneqq q_{\boldsymbol{\phi}}(\boldsymbol{z}|D_c)$, then $\mathcal{L}_{\text{NP}}$ may be thought of as performing VI in this *collection* of models. However, there is no guarantee that these conditional priors are consistent in the sense that they correspond to a single Bayesian model as in Eq (4).

For the non-discretized ConvNP, Eq (5) involves KL divergences between SPs which cannot be computed directly and must be treated carefully [24, 36]. On the other hand, for the discretized ConvNP, the KL divergences can be computed, but grow in magnitude as the discretization becomes finer, and it is not clear that the KL divergence between SPs is recovered in the limit. This raises practical issues for the use of Eq (5) with the ConvNP, as the balance between the two terms depends on the choice of discretization.

## 4.2 Maximum-Likelihood vs Variational Lower Bound Maximization for Training NPs

We argue that the VI interpretation is unnecessary when focusing on predictive performance, and particularly detrimental for ConvNPs, where $\boldsymbol{z}$ has many elements. Noting the equivalence

$$\mathcal{L}_{\text{NP}}(\boldsymbol{\theta}, \boldsymbol{\phi}; \xi) = \mathcal{L}_{\text{ML}}(\boldsymbol{\theta}, \boldsymbol{\phi}; \xi) - \text{KL}\left(q_{\boldsymbol{\phi}}(\boldsymbol{z}|D_c \cup D_t)\|p_{\boldsymbol{\theta}}(D_t|\boldsymbol{z})q_{\boldsymbol{\phi}}(\boldsymbol{z}|D_c)/Z\right), \tag{6}$$

Table 1: Log-likelihoods on 1D regression tasks. Lower bounds marked with asterisk. Highest non-GP values in bold.

| | | WITHIN TRAINING RANGE | | | BEYOND TRAINING RANGE | | |
|---|---|---|---|---|---|---|---|
| | | Matérn-$\frac{5}{2}$ | Weakly Per. | Sawtooth | Matérn-$\frac{5}{2}$ | Weakly Per. | Sawtooth |
| GP | (full) | $1.22_{\pm 6\text{E}-3}$ | $-0.06_{\pm 5\text{E}-3}$ | N/A | $1.22_{\pm 6\text{E}-3}$ | $-0.06_{\pm 5\text{E}-3}$ | N/A |
| ConvNP* | ($\mathcal{L}_{\mathrm{ML}}$) | $\mathbf{-0.58}_{\pm 0.01}$ | $\mathbf{-1.02}_{\pm 6\text{E}-3}$ | $\mathbf{2.30}_{\pm 0.01}$ | $\mathbf{-0.58}_{\pm 0.01}$ | $\mathbf{-1.03}_{\pm 6\text{E}-3}$ | $\mathbf{2.29}_{\pm 0.02}$ |
| ANP* | ($\mathcal{L}_{\mathrm{ML}}$) | $-0.73_{\pm 0.01}$ | $-1.14_{\pm 6\text{E}-3}$ | $0.09_{\pm 3\text{E}-3}$ | $-1.39_{\pm 7\text{E}-3}$ | $-1.35_{\pm 4\text{E}-3}$ | $-0.17_{\pm 1\text{E}-3}$ |
| ANP* | ($\mathcal{L}_{\mathrm{NP}}$) | $-0.96_{\pm 0.01}$ | $-1.37_{\pm 6\text{E}-3}$ | $0.20_{\pm 9\text{E}-3}$ | $-1.48_{\pm 4\text{E}-3}$ | $-1.66_{\pm 0.01}$ | $-0.30_{\pm 4\text{E}-3}$ |
| GP | (diag) | $-0.84_{\pm 9\text{E}-3}$ | $-1.17_{\pm 5\text{E}-3}$ | N/A | $-0.84_{\pm 9\text{E}-3}$ | $-1.17_{\pm 5\text{E}-3}$ | N/A |
| ConvCNP | | $-0.88_{\pm 0.01}$ | $-1.19_{\pm 7\text{E}-3}$ | $1.15_{\pm 0.04}$ | $-0.87_{\pm 0.01}$ | $-1.19_{\pm 7\text{E}-3}$ | $1.11_{\pm 0.04}$ |

where $Z$ is a normalizing constant (see App F for a full derivation), we see that $\mathcal{L}_{\mathrm{NP}}$ is equal to $\mathcal{L}_{\mathrm{ML}}$ up to an additional KL term. This KL term encourages consistency among the $q_\phi(z|D)$ in the sense that Bayes' theorem is respected if the target set is subsumed into the context set. In the infinite capacity/data limit, $\mathcal{L}_{\mathrm{NP}}$ is globally maximized if the ConvNP recovers (i) the prediction map $\pi_P$ for $y_t$ and (ii) exact inference for $z$. This follows from (i) Prop 1, since $\pi_P$ globally optimizes $\mathcal{L}_{\mathrm{ML}}$; and (ii) that exact inference for $z$ is Bayes-consistent, sending the KL term to zero. In most applications, only the distribution over $y_t$ is of interest. Given only finite capacity/data, it can be advantageous to not expend capacity in enforcing Bayes-consistency for $z$, which suggests it could be beneficial to use $\mathcal{L}_{\mathrm{ML}}$ over $\mathcal{L}_{\mathrm{NP}}$. Further, $\mathcal{L}_{\mathrm{ML}}$ has the advantage of being easy to specify for any map parameterizing a predictive process, posing no conceptual issues for the ConvNP. In Sec 5 we find that $\mathcal{L}_{\mathrm{ML}}$ significantly outperforms $\mathcal{L}_{\mathrm{NP}}$ for ConvNPs, and often also for ANPs.

## 5 Experiments

We evaluate ConvNPs on a broad range of tasks. Our main questions are: (i) Does the ConvNP produce coherent, meaningful predictive samples? (ii) Can it leverage translation equivariance to outperform baseline methods within and beyond the training range (generalization)? (iii) Does it learn expressive non-Gaussian predictive distributions?

**Evaluation and baselines.** We use several approaches for evaluating NPs. First, as in [11, 14], we provide qualitative comparisons of samples. These allow us to see if the models display meaningful structure, quantify uncertainty, and are able to generalize spatially. Second, NPs lack closed-form likelihoods, so we evaluate *lower bounds* on their predictive log-likelihoods via importance sampling [20]. As these bounds can be quite loose (App G.1), they are primarily useful to show when NPs outperform baselines with *exact* likelihoods, such as GPs and ConvCNPs. Finally, in Sec 5.3 we consider Bayesian optimization to evaluate the usefulness of ConvNPs for downstream tasks. In Secs 5.1 and 5.2, we compare against the Attentive NP (ANP; [14]), which in prior work is trained with $\mathcal{L}_{\mathrm{NP}}$. The ANP architectures used here are comparable to those in Kim et al. [14], and have a parameter count comparable to or greater than the ConvNP. Full details provided in the supplement.[4]

### 5.1 1D Regression

We train on samples from (i) a Matérn-$\frac{5}{2}$ GP, (ii) a weakly periodic GP, and (iii) a non-Gaussian sawtooth process with random shifts and frequency (see App H for details). Fig 3 shows predictive samples, where during training the models only observe data within the grey regions (training range). While samples from the ANP exhibit unnatural "kinks" and do not resemble the underlying process, the ConvNP produces smooth samples for Matérn–$\frac{5}{2}$ and samples exhibiting meaningful structure for the weakly periodic and sawtooth processes. The ConvNP also generalizes gracefully beyond the training range, whereas ANP fails catastrophically. The ANP with $\mathcal{L}_{\mathrm{NP}}$ collapses to deterministic samples, with the epistemic uncertainty explained using the heteroskedastic noise $\sigma_y^2(x, z)$. This was also noted in Le et al. [20]. This behaviour is alleviated when training with $\mathcal{L}_{\mathrm{ML}}$, with much of the predictive uncertainty due to variations in the sampled functions.

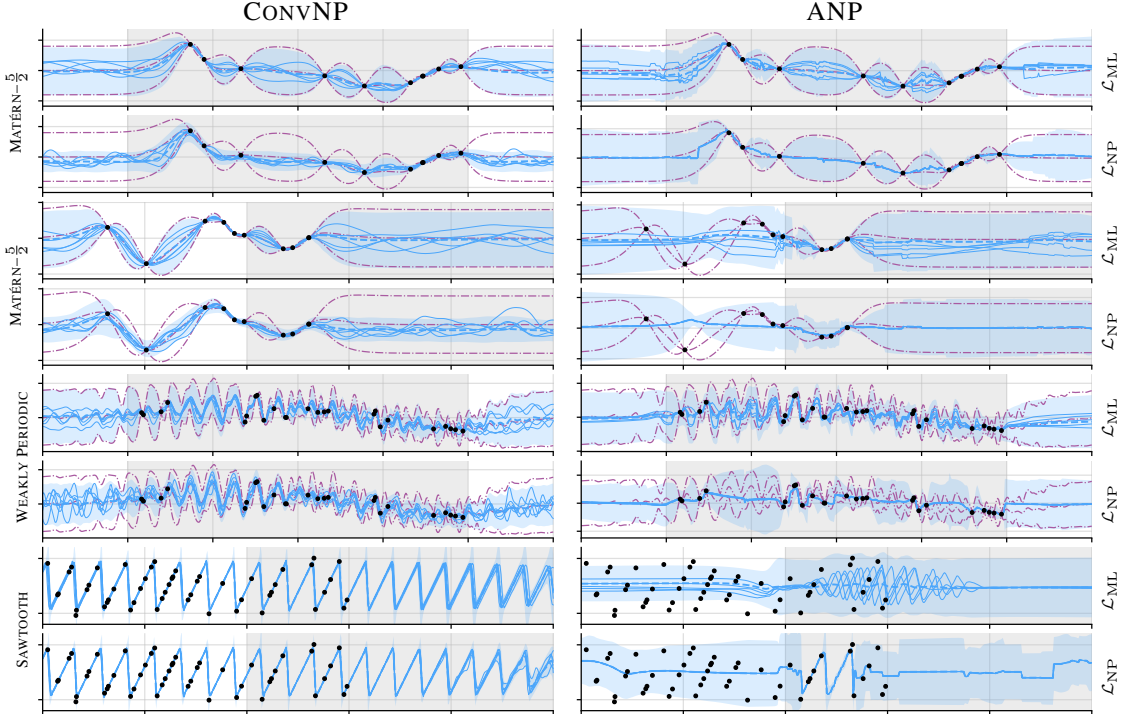

Figure 3: Predictions of ConvNPs and ANPs trained with $\mathcal{L}_{\mathrm{ML}}$ and $\mathcal{L}_{\mathrm{NP}}$, showing interpolation and extrapolation within (grey background) and outside (white background) the training range. Solid blue lines are samples, dashed blue lines are means, and the shaded blue area is $\mu \pm 2\sigma$. Purple dash–dot lines are the ground-truth GP mean and $\mu \pm 2\sigma$. ConvNP handles points outside the training range naturally, whereas this leads to catastrophic failure for the ANP. Note ANP with $\mathcal{L}_{\mathrm{NP}}$ tends to collapse to deterministic samples, with all uncertainty explained with the heteroskedastic noise. In contrast, models trained with $\mathcal{L}_{\mathrm{ML}}$ show diverse samples that account for much of the uncertainty.

Table 2: Test log-likelihood lower bounds for image completion (5 runs).

| | MNIST | | CelebA32 | | SVHN | | ZSMM | |
| | $\mathcal{L}_{\mathrm{ML}}$ | $\mathcal{L}_{\mathrm{NP}}$ | $\mathcal{L}_{\mathrm{ML}}$ | $\mathcal{L}_{\mathrm{NP}}$ | $\mathcal{L}_{\mathrm{ML}}$ | $\mathcal{L}_{\mathrm{NP}}$ | $\mathcal{L}_{\mathrm{ML}}$ | $\mathcal{L}_{\mathrm{NP}}$ |
|---|---|---|---|---|---|---|---|---|
| ConvNP | $\mathbf{2.11}_{\pm 0.01}$ | $0.99_{\pm 0.42}$ | $\mathbf{6.92}_{\pm 0.10}$ | $-0.27_{\pm 0.00}$ | $\mathbf{9.89}_{\pm 0.09}$ | $0.17_{\pm 0.00}$ | $\mathbf{4.58}_{\pm 0.04}$ | $0.14_{\pm 0.00}$ |
| ANP | $1.66_{\pm 0.03}$ | $1.64_{\pm 0.03}$ | $5.98_{\pm 0.08}$ | $6.04_{\pm 0.10}$ | $9.18_{\pm 0.08}$ | $8.91_{\pm 0.06}$ | $-10.8_{\pm 1.99}$ | $-6.45_{\pm 0.99}$ |

Tab 1 compares lower bounds on the log-likelihood for ConvNP with our proposed $\mathcal{L}_{\mathrm{ML}}$ objective and ANP with both $\mathcal{L}_{\mathrm{ML}}$ and the standard $\mathcal{L}_{\mathrm{NP}}$ objective. We also show three *exact* log-likelihoods: (i) the ground-truth GP (full) (ii) the ground-truth GP with diagonalised predictions (diag), and (iii) ConvCNP. The ConvCNP performs on par with GP (diag), which is the optimal factorized predictive. The ConvNP lower bound is consistently higher than the GP (diag) and ConvCNP log-likelihoods, demonstrating that its correlated predictives improve predictive performance. Further, the ConvNP performs similarly inside and outside its training range, demonstrating that TE helps generalization; this is in contrast to the ANP, which fails catastrophically outside its training range. In App I, we provide a thorough comparison for multiple models, training objectives, and data sets.

## 5.2 Image Completion

We evaluate ConvNPs on image completion tasks focusing on spatial generalization. To test this, we consider zero-shot multi MNIST (ZSMM), where we train on single MNIST digits but test on two MNIST digits on a larger canvas. We randomly translate the digits during training, so the generative SP is stationary. The black background on MNIST causes difficulty with heteroskedastic noise, as the models can obtain high likelihood by predicting the background with high confidence whilst ignoring the digits. Hence for MNIST and ZSMM we use homoskedastic noise $\sigma_y^2(z)$. Figs 4a and 4b show that the ANP fails to generalize spatially, whereas this is naturally handled by the ConvNP.

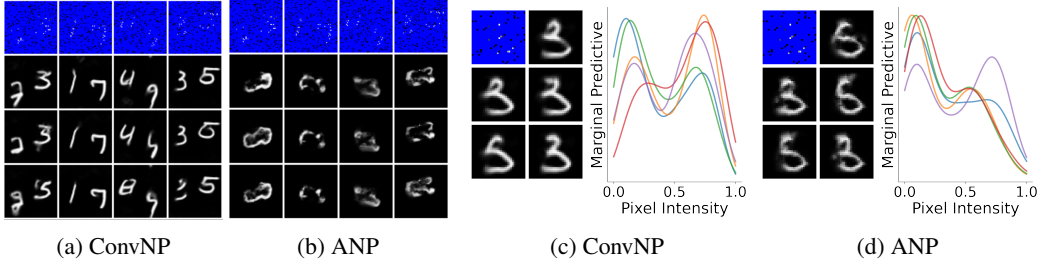

| (a) ConvNP | (b) ANP | (c) ConvNP | (d) ANP |

Figure 4: Left two plots: predictive samples on zero-shot multi MNIST. Right two plots: samples and marginal predictives on standard MNIST. We plot the density of the five marginals that maximize Sarle's bimodality coefficient [9]. We use $\mathcal{L}_{\mathrm{ML}}$ for training. Blue pixels are not in the context set.

Table 3: Joint predictive log-likelihoods (LL) and RMSEs on ERA5-Land, averaged over 1000 tasks.

|  |  | Central (train) | West (test) | East (test) | South (test) |
|---|---|---|---|---|---|
| LL | ConvNP | $\mathbf{4.47}_{\pm 0.07}$ | $\mathbf{4.55}_{\pm 0.08}$ | $\mathbf{5.07}_{\pm 0.07}$ | $\mathbf{4.65}_{\pm 0.08}$ |
|  | GP | $3.33_{\pm 0.06}$ | $3.65_{\pm 0.06}$ | $4.07_{\pm 0.06}$ | $3.34_{\pm 0.06}$ |
| RMSE ($\times 10^{-2}$) | ConvNP | $\mathbf{5.72}_{\pm 0.33}$ | $\mathbf{5.77}_{\pm 0.37}$ | $3.23_{\pm 0.22}$ | $\mathbf{6.92}_{\pm 0.39}$ |
|  | GP | $6.26_{\pm 0.30}$ | $5.75_{\pm 0.29}$ | $\mathbf{3.10}_{\pm 0.18}$ | $7.94_{\pm 0.44}$ |

We also test the ConvNP's ability to learn non-Gaussian predictive distributions. Fig 4c shows that the ConvNP can learn highly multimodal predictives, enabling the generation of diverse yet coherent samples. A quantitative comparison of models using log-likelihood lower bounds is provided in Tab 2, where ConvNP trained with $\mathcal{L}_{\mathrm{ML}}$ consistently achieves the highest values. App J provides details regarding the data, architectures, and protocols used in our image experiments. In App K, we provide samples and further quantitative comparisons of models trained on SVHN [25], MNIST [21], and $32 \times 32$ CelebA [25] in a range of scenarios, along with full experimental details.

### 5.3 Environmental Data

We next consider a real-world data set, ERA5-Land [35], containing environmental measurements at a $\sim$9 km spacing across the globe. We consider predicting daily precipitation $y$ at position $\boldsymbol{x}$. We also provide the model with orography (elevation) and temperature values. We choose a large region of central Europe as our train set, and use regions east, west and south as held-out test sets. For such tasks, models must be able to make predictions at locations spanning a range different from the training set, inhibiting the deployment of NPs not equipped with TE. To sample a task at train time, we sample a random date between 1981 and 2020, then sample a sub-region within the train region, which is split into context and target sets. In this section, we train using $\mathcal{L}_{\mathrm{ML}}$. See App L for details.

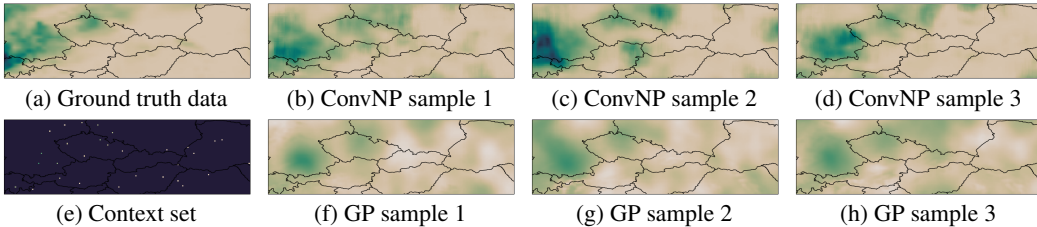

| (a) Ground truth data | (b) ConvNP sample 1 | (c) ConvNP sample 2 | (d) ConvNP sample 3 |
| (e) Context set | (f) GP sample 1 | (g) GP sample 2 | (h) GP sample 3 |

Figure 5: Predictive samples overlaid on central Europe. Darker colours show higher precipitation. In (e), coloured pixels represent context points. GP samples often take negative values (lighter than ground truth data, see App L.2 for a discussion), whereas the NP has learned to produce non-negative samples which capture the *sparsity* of precipitation. The model is trained on subregions roughly the size of the lengthscale of the precipitation process. More samples in App M.

**Prediction.** We first evaluate the ConvNP's predictive performance, comparing to a GP trained individually on each task as a baseline. In about 10% of tasks, the GP obtains a poor likelihood ($< 0$

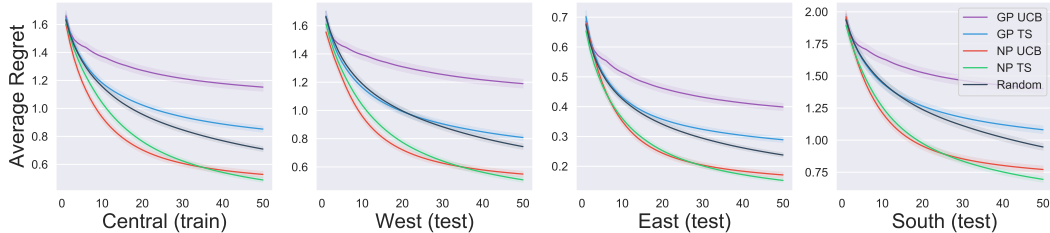

Figure 6: Average regret plotted against number of points queried, averaged over 5000 tasks.

nats); we remove these outliers from the evaluation. The results are shown in Tab 3. The ConvNP and GP have comparable RMSEs except on south, where the ConvNP outperforms the GP. However, the ConvNP consistently outperforms the GP in log-likelihood, which is expected for the following reasons: (i) the GP does not share information between tasks and hence is prone to overfitting on small context sets, resulting in overconfident predictions; and (ii) the ConvNP can learn non-Gaussian predictive densities (illustrated in App M). Fig 5 shows samples from the predictive process of a ConvNP and GP, over the whole of the train region. This demonstrates spatial extrapolation, as the ConvNP is trained only on random subregions.

**Bayesian optimization.** We demonstrate the ConvNP in a downstream task by considering a toy Bayesian optimisation problem, where the goal is to identify the location with heaviest rainfall on a given day. We also test the ConvNP's spatial generalization, by optimising over larger regions (for central, west, and south) than the model was trained on. We test both Thompson sampling (TS) [38] and upper confidence bounds (UCB) [1] as methods for acquiring points. Note that TS requires coherent samples. The results are shown in Fig 6. On all data sets, ConvNP TS and UCB significantly outperform the random baseline by the 50th iteration; the GP does not reliably outperform random. We hypothesize this is due to its overconfidence, in line with the results on prediction.

## 6 Related Work and Discussion

We have introduced the ConvNP, a TE map from observed data sets to predictive SPs. Within the NP framework, ConvNPs bring together three key considerations.

**Expressive joint densities.** ConvNPs extend ConvCNPs to allow for expressive joint predictive densities. A powerful alternative approach is to combine *autoregressive* (AR) models (such as PixelCNN++ [33] and the Image Transformer [26]) with CNPs. A difficulty in introducing AR sampling to CNPs is the need to specify a sampling ordering, which is in tension with permutation invariance and relates to the discussion on Bayes-consistency (Sec 4.2). Several works have considered *exchangeable* NP models [23, 19, 18], providing an avenue for future investigation.

**Translation equivariance.** There has been much interest in incorporating equivariance with respect to symmetry groups into neural networks, e.g. [16, 4, 5, 17], with a comprehensive treatment provided by Bloem-Reddy and Teh [2]. ConvNPs leverage a simple relationship between translation equivariance and stationarity to construct a model particularly well suited to stationary SPs. Similar ideas have been explored for 3D point-cloud modelling [27, 28]. For example, the models proposed in [42, 41] perform convolutions over continuous domains, which are both TE and permutation invariant, achieving excellent performance in point-cloud classification. In contrast with ConvNPs, point-cloud models (i) are generally used as classification function approximators, rather than meta or few-shot learners; (ii) are typically tailored towards point clouds, making heavy use of specific properties for function design; and (iii) have not considered latent variable or stochastic generalizations.

**Neural Process training procedures.** One of the key benefits of CNPs is their simple maximum-likelihood training procedure [10, 13]. In contrast, NPs are usually trained with VI-inspired objectives [11], variants of which are empirically investigated in Le et al. [20]. We propose an alternative training procedure that discards VI in favor of a (biased) maximum-likelihood approach that focuses on directly optimizing predictive performance. In this regard, our work is similar to Gordon et al. [12], albeit in a very different domain. This approach has two benefits: (i) it does not require carefully designed inference procedures, and works "out-of-the-box" for a range of models; and (ii) empirically, we find that it leads to improved performance for ConvNPs and, often, for ANPs.

## Broader Impact

The proposed model and training procedure are geared towards off-the-grid, spatio-temporal applications. As such, ConvNPs are particularly well-suited for many important applications in the medical and environmental sciences, such as modelling electronic healthcare records or the temporal evolution of temperatures. We hope that one impact of ConvNPs is to increase the usability of deep learning tools in the sciences. Another potential application of ConvNPs is image generation, which has potentially negative societal impacts. However, ConvNPs focus on predicting distributions over images, and are far from state-of-the-art in terms of perceptual quality. Thus we believe the societal impact of ConvNPs via image-generation will be insignificant.

## Acknowledgements

The authors would like to thank Invenia Labs for their support during the project. We thank William Tebbutt for insightful discussions. We thank David R. Burt, Eric Nalisnick, Cozmin Ududec and John Bronskill for helpful comments on the manuscript. Andrew Y. K. Foong gratefully acknowledges funding from a Trinity Hall Research Studentship and the George and Lilian Schiff Foundation. Part of the work was done while Yann Dubois was working as an AI resident at Facebook. Richard E. Turner is supported by Google, Amazon, ARM, Improbable, EPSRC grants EP/M0269571 and EP/L000776/1, and the UKRI Centre for Doctoral Training in the Application of Artificial Intelligence to the study of Environmental Risks (AI4ER).

## Footnotes

[2]i.e., $(d_{\boldsymbol{\theta}})_*(\mathrm{E}_{\boldsymbol{\phi}})$ is the measure induced on $\mathbb{R}^{\mathcal{X}}$ by sampling a function from $\mathrm{E}_{\boldsymbol{\phi}}$ and passing it through $d_{\boldsymbol{\theta}}$.

[3] See Gordon et al. [13, Appendix D.6] for a discussion.

[4]Code to reproduce the 1D regression experiments can be found at `https://github.com/wesselb/NeuralProcesses.jl`, and code to implement the image-completion experiments can be found at `https://github.com/YannDubs/Neural-Process-Family`.

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
