[Supplementary Material]

# Supplementary Material: Meta-Learning Stationary Stochastic Process Prediction with Convolutional Neural Processes

**Andrew Y. K. Foong**[*]
University of Cambridge
ykf21@cam.ac.uk

**Wessel P. Bruinsma**[*]
University of Cambridge
Invenia Labs
wpb23@cam.ac.uk

**Jonathan Gordon**[*]
University of Cambridge
jg801@cam.ac.uk

**Yann Dubois**
Facebook AI Research
yannd@fb.com

**James Requeima**
University of Cambridge
Invenia Labs
jrr41@cam.ac.uk

**Richard E. Turner**
University of Cambridge
Microsoft Research
ret26@cam.ac.uk

## A  Formal Definitions and Set-up

**Notation.** We first review the notation introduced in the main body for convenience. Let $\mathcal{X} = \mathbb{R}^{d_{\text{in}}}$ and $\mathcal{Y} = \mathbb{R}$ denote the input and output spaces respectively, and let $(x, y)$ denote a generic input-output pair (higher-dimensional outputs can be treated easily). Define $\mathcal{S}_N = (\mathcal{X} \times \mathcal{Y})^N$ to be the collection of all data sets of size $N$, and let $\mathcal{S} := \bigcup_{N=1}^{\infty} \mathcal{S}_N$. Let $D_c, D_t \in \mathcal{S}$ denote a *context* and *target* set respectively. Later, as is common in recent meta-learning approaches, we will consider predicting the target set from the context set Garnelo et al. [3, 4]. Let $\boldsymbol{X}_c = (\boldsymbol{x}_1, \ldots, \boldsymbol{x}_{N_c})$ denote a matrix of context set inputs, with $\boldsymbol{y}_c = (y_1, \ldots, y_{N_c})$ the corresponding outputs; $\boldsymbol{X}_t, \boldsymbol{y}_t$ are defined analogously. We denote a single *task* as $\xi = (D_c, D_t) = (D_c, (\boldsymbol{X}_t, \boldsymbol{y}_t))$.

**Stochastic processes.** For our purposes, a stochastic process on $\mathcal{X}$ will be defined[2] as a probability measure on the set of functions from $\mathcal{X} \to \mathbb{R}$, i.e. $\mathbb{R}^{\mathcal{X}}$, equipped with the product $\sigma$-algebra of the Borel $\sigma$-algebra over each index point [16], denoted $\Sigma$. The measurable sets of $\Sigma$ are those which can be specified by the values of the function at a countable subset $I \subset \mathcal{X}$ of its input locations. Since in practice we only ever observe data at a finite number of points, this is sufficient for our purposes. We denote the set of all such measures as $\mathcal{P}(\mathcal{X})$. We model the world as having a ground truth stochastic process $P \in \mathcal{P}(\mathcal{X})$. Consider a Kolmogorov-consistent (i.e. consistent under marginalization) collection of distributions on finite index sets $I \subset \mathcal{X}$. By the Kolmogorov extension theorem, there exists a unique measure on $(\mathbb{R}^{\mathcal{X}}, \Sigma)$ that has these distributions as its finite marginals. Hence we may think of these stochastic processes as defined by their finite-dimensional marginals.

**Conditioning on observations.** We now define what it means to condition on observations of the stochastic process $P$. Let $p(\boldsymbol{y}|\boldsymbol{X})$ denote the density with respect to Lebesgue measure of the finite marginal of $P$ with index set $\boldsymbol{X}$ (we assume these densities always exist). Assume we have observed $P$ at a finite number of points $(\boldsymbol{X}_c, \boldsymbol{y}_c)$, with $p(\boldsymbol{y}_c|\boldsymbol{X}_c) > 0$. Let $\boldsymbol{X}_t$ be another finite index set. Then we define the finite marginal at $\boldsymbol{X}_t$ conditioned on $D_c$ as the distribution with density

$$p(\boldsymbol{y}_t|\boldsymbol{X}_t, D_c) = \frac{p(\boldsymbol{y}_t, \boldsymbol{y}_c|\boldsymbol{X}_t, \boldsymbol{X}_c)}{p(\boldsymbol{y}_c|\boldsymbol{X}_c)}. \tag{1}$$

It can easily be verified that for a fixed $D_c$, the conditional marginal distributions for different $\boldsymbol{X}_t$ in Eq (1) are Kolmogorov-consistent. Again, the Kolmogorov extension theorem implies there is a

---

[*]Authors contributed equally.

[2]Strictly speaking, this is non-standard terminology, since $P$ is the *law* of a stochastic process.

unique measure $P_{D_c}$ on $(\mathbb{R}^{\mathcal{X}}, \Sigma)$ that has Eq (1) as its finite marginals. We now define $\pi_P : \mathcal{S} \to \mathcal{P}(\mathcal{X}), \pi_P : D_c \mapsto P_{D_c}$ as the *prediction map*, so called because it maps each observed dataset $D_c$ to the exact predictive stochastic process conditioned on $D_c$. The meta-learning task may be viewed as learning an approximation to the prediction map.

## B  Stationary Processes and Translation Equivariance

**Def 1** (Translating data sets and SPs). We define the action of the translation operator $T_{\boldsymbol{\tau}}$ on data sets and SPs, where $\boldsymbol{\tau} \in \mathcal{X}$ denotes the shift vector of the translation.[3]

1. Let $(\boldsymbol{x}_n, \boldsymbol{y}_n)_{n=1}^N = S \in \mathcal{S}$. For the index set $\boldsymbol{X} = (\boldsymbol{x}_1, \dots, \boldsymbol{x}_n)$, the translation by $\boldsymbol{\tau}$ is defined as $T_{\boldsymbol{\tau}} \boldsymbol{X} = (\boldsymbol{x}_1 + \boldsymbol{\tau}, \dots, \boldsymbol{x}_n + \boldsymbol{\tau})$. Similarly, $T_{\boldsymbol{\tau}} S := (\boldsymbol{x}_n + \boldsymbol{\tau}, \boldsymbol{y}_n)_{n=1}^N$.
2. For a function $f \in \mathbb{R}^{\mathcal{X}}$, define $T_{\boldsymbol{\tau}} f(\boldsymbol{x}) := f(\boldsymbol{x} - \boldsymbol{\tau})$ for all $\boldsymbol{x} \in \mathcal{X}$. Let $F \in \Sigma$ be a measurable set of functions. Then $T_{\boldsymbol{\tau}} F := \{T_{\boldsymbol{\tau}} f : f \in F\}$.
3. For any SP $P \in \mathcal{P}(\mathcal{X})$, we now define $T_{\boldsymbol{\tau}} P$ by setting[4] $T_{\boldsymbol{\tau}} P(F) := P(T_{-\boldsymbol{\tau}} F)$ for all $F \in \Sigma$.

**Def 2** (Stationary SP). We say a stochastic process is (strictly) *stationary* if the densities of its finite marginals satisfy

$$p(\boldsymbol{y}_t | \boldsymbol{X}_t) = p(\boldsymbol{y}_t | T_{\boldsymbol{\tau}} \boldsymbol{X}_t) \tag{2}$$

for all $\boldsymbol{y}_t$, $\boldsymbol{X}_t$ and $\boldsymbol{\tau}$.

**Def 3** (Translation equivariant prediction maps). We say that $\Psi : \mathcal{S} \to \mathcal{P}(\mathcal{X})$ is *translation equivariant* if $\Psi(T_{\boldsymbol{\tau}} S) = T_{\boldsymbol{\tau}} \Psi(S)$ for any data set $S \in \mathcal{S}$ and shift $\boldsymbol{\tau} \in \mathcal{X}$.

The following simple statement highlights the link between stationarity and translation equivariance:

**Prop 1.** Let $P$ be a stationary SP. Then the prediction map $\pi_P$ is translation equivariant.[5]

*Proof.* Let $p(\boldsymbol{y}_t | \boldsymbol{X}_t, D_c)$ denote the finite dimensional density of $\pi_P(D_c)$ at index set $\boldsymbol{X}_t$. To show that $\pi_P(T_{\boldsymbol{\tau}} D_c) = T_{\boldsymbol{\tau}} \pi_P(D_c)$ it suffices to show that $p(\boldsymbol{y}_t | \boldsymbol{X}_t, T_{\boldsymbol{\tau}} D_c) = p(\boldsymbol{y}_t | T_{-\boldsymbol{\tau}} \boldsymbol{X}_t, D_c)$. We have

$$
\begin{aligned}
p(\boldsymbol{y}_t | \boldsymbol{X}_t, T_{\boldsymbol{\tau}} D_c) &= \frac{p(\boldsymbol{y}_t, \boldsymbol{y}_c | \boldsymbol{X}_t, T_{\boldsymbol{\tau}} \boldsymbol{X}_c)}{p(\boldsymbol{y}_c | T_{\boldsymbol{\tau}} \boldsymbol{X}_c)} \\
&= \frac{p(\boldsymbol{y}_t, \boldsymbol{y}_c | T_{-\boldsymbol{\tau}} \boldsymbol{X}_t, \boldsymbol{X}_c)}{p(\boldsymbol{y}_c | \boldsymbol{X}_c)} \\
&= p(\boldsymbol{y}_t | T_{-\boldsymbol{\tau}} \boldsymbol{X}_t, D_c),
\end{aligned}
$$

where we used the stationarity assumption in the second line. □

## C  Description and Pseudocode for ConvCNP and ConvNP

We provide additional details and pseudo-code for ConvCNP and ConvNP. Similar to Gordon et al. [5], we distinguish between the "on-the-grid" and "off-the-grid" versions of the model. In our experiments, we use the "off-the-grid" version of the model for the 1d experiments in Sec 5.1, and the "on-the-grid" version for the image and environmental experiments in Secs 5.2 and 5.3.

### C.1  ConvCNP Pseudo-Code and Details

**Off-the-grid ConvCNP.** We begin by providing details for off-the-grid ConvCNP. As detailed in the main text, the encoder $\mathrm{E}_\phi$ is defined by a ConvCNP, which provides a distribution over latent functions $z$. In practice, we consider the discretized version, where we denote the grid of discretization locations as $(\boldsymbol{t}_i)_{i=1}^K$, with $\boldsymbol{t}_i \in \mathcal{X}$. Let $p_\phi(\boldsymbol{z}_i | \boldsymbol{t}_i, D_c)$ denote the density of the latent function at the

**Algorithm 1** Forward pass through ConvCNP (off-the-grid)

---

**Require:** $\rho = (\text{CNN}, \psi_\rho)$, $\psi$, and density $\zeta$
**Require:** context $(\boldsymbol{x}_n, y_n)_{n=1}^N$, target $(\boldsymbol{x}_m^*)_{m=1}^M$
 1: lower, upper $\leftarrow$ range$\left((\boldsymbol{x}_n)_{n=1}^N \cup (\boldsymbol{x}_m^*)_{m=1}^M\right)$
 2: $(\boldsymbol{t}_i)_{i=1}^K \leftarrow$ uniform_grid(lower, upper; $\gamma$)
 3: $\boldsymbol{h}_i \leftarrow \sum_{n=1}^N \begin{bmatrix} 1 & y_n \end{bmatrix}^\top \psi(\boldsymbol{t}_i - \boldsymbol{x}_n)$
 4: $\boldsymbol{h}_i^{(1)} \leftarrow \boldsymbol{h}_i^{(1)} / \boldsymbol{h}_i^{(0)}$
 5: $(f_\mu(\boldsymbol{t}_i), f_\sigma(\boldsymbol{t}_i))_{i=1}^T \leftarrow \text{CNN}((\boldsymbol{t}_i, \boldsymbol{h}_i)_{i=1}^T)$
 6: $\boldsymbol{\mu}_m \leftarrow \sum_{i=1}^K f_\mu(\boldsymbol{t}_i)\psi_\rho(\boldsymbol{x}_m^* - \boldsymbol{t}_i)$
 7: $\boldsymbol{\sigma}_m \leftarrow \sum_{i=1}^K \text{pos}(f_\sigma(\boldsymbol{t}_i))\psi_\rho(\boldsymbol{x}_m^* - \boldsymbol{t}_i)$
 8: **return** $(\boldsymbol{\mu}_m, \boldsymbol{\sigma}_m)_{m=1}^M$

---

$i$th position, i.e. at $\boldsymbol{z}_i = z(\boldsymbol{t}_i)$. Then in order to sample $\boldsymbol{z} \sim \text{E}_\phi$ (as in e.g. Eq (2) in the main body) we specify the density of the entire discretized latent function $\boldsymbol{z}$ as:

$$p_\phi(\boldsymbol{z}|D_c) = \prod_{i=1}^K p_\phi(\boldsymbol{z}_i|\boldsymbol{t}_i, D_c) = \prod_{i=1}^K \mathcal{N}(\boldsymbol{z}_i; \mu(\boldsymbol{t}_i, D_c), \sigma^2(\boldsymbol{t}_i, D_c)), \tag{3}$$

where $\mu$ and $\sigma^2$ are parametrized by ConvDeepSets [5].

ConvDeepSets can be expressed as the composition of two functions. Let $\Phi = \rho \circ \gamma$ be a ConvDeepSet. $\gamma$ maps a data set $D$ to its functional representation via

$$\gamma(D) = \sum_{(\boldsymbol{x}, y) \in D} \phi(y)\psi(\cdot - \boldsymbol{x}).$$

Following Gordon et al. [5], we set $\phi(y) = [1, y]^\top \in \mathbb{R}^2$, and $\psi$ to be a radial basis function. $\gamma(D)$ is itself discretized by evaluating it on a grid (which for simplicity we can also take to be $(\boldsymbol{t}_i)_{i=1}^K$).

Next, $\rho$ maps the discretized $\gamma(D)$ to a continuous function, which we denote $f = \rho(\gamma(D))$. $\gamma$ is itself implemented in two stages. First a deep CNN maps the discretized $\gamma(D)$ to a discretized output. Second, this discrete output is mapped to a continuous function by using the CNN outputs as weights for evenly-spaced basis functions (again employing radial basis functions), which we denote by $\psi_\rho$.

Whenever models output standard deviations, we enforce positivity via a function (e.g. the soft-plus function), which we denote $\text{pos}(\cdot)$. Pseudo-code for a forward pass through an off-the-grid ConvCNP is provided in Algorithm 1. Note the forward pass involves the computation of a *density channel* $\boldsymbol{h}^{(0)}$, whose role intuitively is to allow the model to know where it has observed datapoints. This is discussed further in Gordon et al. [5].

**On-the-grid ConvCNP.** Next, we describe the ConvCNP for on-the-grid data, which is used in our image and environmental experiments. This version is simpler to implement in practice, and is applicable whenever the input data is confined to a regular grid. As in Gordon et al. [5] we choose the discretization $(\boldsymbol{t}_i)_{i=1}^K$ to be the pixel locations.

Let $\text{I} \in \mathbb{R}^{H \times W \times C}$ be an image of dimensions $H, W, C$ (height, width, and channels, respectively). We define a mask $\text{M}_c$, which is such that $[\text{M}_c]_{i,j} = 1$ if pixel location $(i, j)$ is in the context set, and 0 otherwise. Masking an image is then achieved via element-wise multiplication, denoted $\text{M}_c \odot \text{I}$. This allows us to flexibly define context and target sets for an image (target sets are typically considered as the complete image, so the masks $\text{M}_c$ are simply binary-valued tensors with the same dimensions as the image). In this setting, we implement $\phi$, by selecting the context points, and prepend the context mask: $\phi = [\text{M}_c, \text{Z}_c]^\top$. We then implement $\gamma$ by a simple convolutional layer, which we denote $\text{CONV}_\theta$ to emphasize that we use a standard 2d convolutional layer. Full pseudo-code for the on-the-grid ConvCNP is provided in Algorithm 2.

### C.2 Pseudo-Code for the ConvNP

The ConvNP can be implemented very simply by passing samples from the ConvCNP through an additional CNN decoder, which we denote $d_\theta$. For an "off-the-grid" ConvNP, similarly to the

---

**Algorithm 2** ConvCNP Forward pass (on-the-grid)

---

**Require:** $\rho = (\text{CNN}, \psi_\rho)$ and $\text{CONV}_{\boldsymbol{\theta}}$
**Require:** image I, context $\text{M}_c$, and target mask $\text{M}_t$
 1: We discretize at the pixel locations.
 2: $\text{I}_c \leftarrow \text{M}_c \odot \text{I}$
 3: $\boldsymbol{h} \leftarrow \text{CONV}_{\boldsymbol{\theta}}([\text{M}_c, \text{I}_c]^\top)$
 4: $\boldsymbol{h}^{(1:C)} \leftarrow \boldsymbol{h}^{(1:C)}/\boldsymbol{h}^{(0)}$
 5: $f_t \leftarrow \text{M}_t \odot \text{CNN}(\boldsymbol{h})$
 6: $\boldsymbol{\mu} \leftarrow f_t^{(1:C)}$
 7: $\boldsymbol{\sigma} \leftarrow \text{pos}(f_t^{(C+1:2C)})$
 8: **return** $(\boldsymbol{\mu}, \boldsymbol{\sigma})$

---

---

**Algorithm 3** Forward pass through ConvNP (off-the-grid)

---

**Require:** $d = (\text{CNN}, \psi_d)$, $\text{E}_{\boldsymbol{\phi}}$ (off-the-grid ConvCNP), and number of samples $L$
**Require:** context $(\boldsymbol{x}_n, y_n)_{n=1}^N$, target $(\boldsymbol{x}_m^*)_{m=1}^M$
 1: $\boldsymbol{\mu}_z, \boldsymbol{\sigma}_z \leftarrow \text{E}_{\boldsymbol{\phi}}(D_c)$
 2: **for** $l = 1, \ldots, L$ **do**
 3: $\quad \boldsymbol{z}_l \sim \mathcal{N}(\boldsymbol{z}; \boldsymbol{\mu}_z, \boldsymbol{\sigma}_z^2)$
 4: $\quad (f_\mu(\boldsymbol{t}_i), f_\sigma(\boldsymbol{t}_i))_{i=1}^K \leftarrow \text{CNN}(\boldsymbol{z}_l)$
 5: $\quad \boldsymbol{\mu}_{m,l} \leftarrow \sum_{i=1}^T f_\mu(\boldsymbol{t}_i)\psi_d(\boldsymbol{x}_m^* - \boldsymbol{t}_i)$
 6: $\quad \boldsymbol{\sigma}_{m,l} \leftarrow \text{pos}(f_\sigma(\boldsymbol{t}_i))$
 7: **end for**
 8: **return** $(\boldsymbol{\mu}, \boldsymbol{\sigma})$

---

ConvCNP, we must map the output of a standard CNN back to functions on a continuous domain $\mathcal{X}$. This can be achieved via an RBF mapping, similar to the off-the-grid ConvCNP, e.g. Algorithm 1 lines 6, 7. Pseudo-code for off- and on-the-grid ConvNPs are provided in Algorithms 3 and 4, respectively. Note that for the ConvNP, the discretization of the latent function $\boldsymbol{z}$ is typically on a pre-specified grid, and therefore lines 6 and 7 of Algorithm 1 are unnecessary when calling the ConvCNP (Algorithm 3, line 1).

## D  Translation Equivariance of the ConvNP

We prove that the ConvNP is a translation equivariant map from data sets to stochastic processes, by proving that the decoder and encoder are separately translation equivariant. In this section we suppress the dependence on parameters $(\boldsymbol{\phi}, \boldsymbol{\theta})$.

**Lem 1.** Let $d$ be a measurable, translation equivariant map from $(\mathbb{R}^{\mathcal{X}}, \Sigma)$ to $(\mathbb{R}^{\mathcal{X}}, \Sigma)$. The ConvNP decoder $\text{D} : \mathcal{P}(\mathcal{X}) \to \mathcal{P}(\mathcal{X})$, defined by $\text{D}(P) = d_*(P)$, where $d_*(P)$ is the pushforward measure under $d$, is translation equivariant.

*Proof.* Let $F \in \Sigma$ be measurable. Then:

$$
\begin{aligned}
\text{D}(T_{\boldsymbol{\tau}} P)(F) &\overset{\text{(a)}}{=} T_{\boldsymbol{\tau}} P(d^{-1}(F)) \\
&= P(T_{-\boldsymbol{\tau}} d^{-1}(F)) \\
&\overset{\text{(b)}}{=} P(d^{-1}(T_{-\boldsymbol{\tau}} F)) \\
&= \text{D}(P)(T_{-\boldsymbol{\tau}} F) \\
&= T_{\boldsymbol{\tau}} \text{D}(P)(F).
\end{aligned}
$$

**Algorithm 4** Forward pass through ConvNP (on-the-grid)

---

**Require:** $d =$ CNN, $E_\phi$ (on-the-grid ConvCNP), and number of samples $L$
**Require:** image I, context mask $M_c$, and target mask $M_t$
  1: $\boldsymbol{\mu}_z, \boldsymbol{\sigma}_z \leftarrow E_\phi(I, M_c)$
  2: **for** $l = 1, \ldots, L$ **do**
  3:      $\boldsymbol{z}_l \sim \mathcal{N}(\boldsymbol{z}; \boldsymbol{\mu}_z, \boldsymbol{\sigma}_z^2)$
  4:      $(f_\mu(\boldsymbol{t}_i), f_\sigma(\boldsymbol{t}_i))_{i=1}^K \leftarrow$ CNN$(\boldsymbol{z}_l)$
  5:      $\boldsymbol{\mu} \leftarrow f_t^{(1:C)}$
  6:      $\boldsymbol{\sigma} \leftarrow \text{pos}\left(f_t^{(C+1:2C)}\right)$
  7: **end for**
  8: **return** $(\boldsymbol{\mu}, \boldsymbol{\sigma})$

---

Here (a) follows from definition of the pushforward, and (b) follows because

$$
\begin{aligned}
T_{-\boldsymbol{\tau}} d^{-1}(F) &= T_{-\boldsymbol{\tau}}\{f : d(f) \in F\} \\
&= \{T_{-\boldsymbol{\tau}} f : d(f) \in F\} \\
&= \{f : d(T_{\boldsymbol{\tau}} f) \in F\} \\
&= \{f : T_{\boldsymbol{\tau}} d(f) \in F\} \\
&= \{f : d(f) \in T_{-\boldsymbol{\tau}} F\} \\
&= d^{-1}(T_{-\boldsymbol{\tau}} F). \qquad \square
\end{aligned}
$$

**Lem 2.** The ConvNP encoder E (a ConvCNP), is a translation equivariant map from data sets to stochastic processes.

*Proof.* Recall that the mean and variance $\mu(\cdot, S), \sigma^2(\cdot, S)$ (viewed as maps from $S \to C_b(\mathcal{X})$) of the encoder E are both given by ConvDeepSets. Due to the translation equivariance of ConvDeepSets [5, Theorem 1], $\mu(\cdot, T_{\boldsymbol{\tau}} S) = T_{\boldsymbol{\tau}} \mu(\cdot, S)$ for all $S, \boldsymbol{\tau}$, and similarly for $\sigma^2$. Let $F \in \Sigma$. Then since the measure $E(S) \in \mathcal{P}_N(\mathcal{X})$ is defined entirely by its mean and variance function, $E(T_{\boldsymbol{\tau}} S)(F) = E(S)(T_{-\boldsymbol{\tau}} F) = T_{\boldsymbol{\tau}} E(S)(F)$. $\qquad \square$

Noting that a composition of translation equivariant maps is itself translation equivariant, we obtain the following proposition:

**Prop 2.** Define ConvNP $=$ D $\circ$ E. Then ConvNP is a translation equivariant map from data sets to stochastic processes.

# E   Recovering the Prediction Map in the Infinite Data / Capacity Limits

**Task generation procedure.** Assume tasks $\xi = (D_c, D_t)$ are generated as follows: first, some finite number of input locations $\boldsymbol{X}_t, \boldsymbol{X}_c$ are sampled. Assume that $\Pr(|\boldsymbol{X}_t| = n) > 0$ for all $n \in \mathbb{Z}_{\geq 0}$, where $|\boldsymbol{X}_t|$ denotes the number of datapoints in $\boldsymbol{X}_t$, and assume the same is true of $\Pr(|\boldsymbol{X}_c| = n)$. Further assume that for each $n > 0$, the distribution of $\boldsymbol{X}$ given $|\boldsymbol{X}| = n$ has a continuous density with support over all of $\mathbb{R}^{n \times d_{\text{in}}}$. Next, we sample $\boldsymbol{y}_t, \boldsymbol{y}_c$ from the finite marginal of the ground truth stochastic process $P$, which has density $p(\boldsymbol{y}_t, \boldsymbol{y}_c | \boldsymbol{X}_t, \boldsymbol{X}_c)$. Finally, we set $(D_c, D_t) \coloneqq ((\boldsymbol{X}_t, \boldsymbol{y}_t), (\boldsymbol{X}_c, \boldsymbol{y}_c))$.

**Prop 3.** Let $\Psi : S \to \mathcal{P}(\mathcal{X})$ be any map from data sets to stochastic processes, and let $\mathcal{L}_{\text{ML}}(\Psi) \coloneqq \mathbb{E}_{p(\xi)}[\log p_\Psi(\boldsymbol{y}_t | \boldsymbol{X}_t, D_c)]$, where the density $p_\Psi$ is that of $\Psi(D_c)$ evaluated at $\boldsymbol{X}_t$. Then $\Psi$ globally maximises $\mathcal{L}_{\text{ML}}$ if and only if $\Psi = \pi_P$, the prediction map.

*Proof.* We have:

$$\mathcal{L}_{\text{ML}}(\Psi) = \mathbb{E}_{p(D_c, \boldsymbol{X}_t, \boldsymbol{y}_t)}\left[\log p_\Psi(\boldsymbol{y}_t | \boldsymbol{X}_t, D_c)\right] \tag{4}$$

$$= \mathbb{E}_{p(D_c, \boldsymbol{X}_t)}\left[\mathbb{E}_{p(\boldsymbol{y}_t | \boldsymbol{X}_t, D_c)}\left[\log p_\Psi(\boldsymbol{y}_t | \boldsymbol{X}_t, D_c)\right]\right] \tag{5}$$

$$= -\mathbb{E}_{p(D_c, \boldsymbol{X}_t)}\left[\text{KL}\left(p(\boldsymbol{y}_t | \boldsymbol{X}_t, D_c)\|p_\Psi(\boldsymbol{y}_t | \boldsymbol{X}_t, D_c)\right)\right] + \text{constant}, \tag{6}$$

where the additive constant is constant with respect to $\Psi$. First note that the KL-divergence is non-negative, and that the prediction map sends all the KL-divergences to zero, globally optimising $\mathcal{L}(\Psi)$. Furthermore, the KL-divergence is equal to zero if and only if the two distributions are equal, and this must hold for all $\boldsymbol{X}_t, D_c$. For, if this were not the case, the KL-divergence would contribute a non-zero amount to the expectation in Eq (6). □

Strictly speaking, this argument only shows that the finite marginals of the prediction map and $\Psi$ must be equal for almost all $(D_c, \boldsymbol{X}_t)$ with respect to $p(D_c, \boldsymbol{X}_t)$. Since the task generation procedure outlined in this section assumes a finite probability of generating any finite-sized context and target set, this is not very restrictive. However, in practice we often limit the maximum size of the sampled data sets, and also their range in $\mathcal{X}$ space. Hence we can only expect the model to learn reasonable predictions within the ranges seen during train time.

# F   Relationship Between Neural Process and Maximum-Likelihood Objectives

Let $D := D_t \cup D_c$, and let $Z = \int p_{\boldsymbol{\theta}}(\boldsymbol{y}_t | \boldsymbol{X}_t, \boldsymbol{z})q_{\boldsymbol{\phi}}(\boldsymbol{z}|D_c)\,\mathrm{d}\boldsymbol{z}$. The NP objective is:

$$\mathcal{L}_{\text{NP}}(\boldsymbol{\theta}, \boldsymbol{\phi}; \xi) := \mathbb{E}_{q_{\boldsymbol{\phi}}(\boldsymbol{z}|D)}[\log p_{\boldsymbol{\theta}}(\boldsymbol{y}_t|\boldsymbol{X}_t, \boldsymbol{z})] - \text{KL}(q_{\boldsymbol{\phi}}(\boldsymbol{z}|D)\|q_{\boldsymbol{\phi}}(\boldsymbol{z}|D_c)) \tag{7}$$

$$= \mathbb{E}_{q_{\boldsymbol{\phi}}(\boldsymbol{z}|D)}[\log p_{\boldsymbol{\theta}}(\boldsymbol{y}_t|\boldsymbol{X}_t, \boldsymbol{z}) + \log q_{\boldsymbol{\phi}}(\boldsymbol{z}|D_c) - \log q_{\boldsymbol{\phi}}(\boldsymbol{z}|D)] \tag{8}$$

$$= \mathbb{E}_{q_{\boldsymbol{\phi}}(\boldsymbol{z}|D)}\left[\log Z + \log \frac{p_{\boldsymbol{\theta}}(\boldsymbol{y}_t|\boldsymbol{X}_t, \boldsymbol{z})q_{\boldsymbol{\phi}}(\boldsymbol{z}|D_c)}{Z} - \log q_{\boldsymbol{\phi}}(\boldsymbol{z}|D)\right] \tag{9}$$

$$= \log Z - \text{KL}\left(q_{\boldsymbol{\phi}}(\boldsymbol{z}|D)\left\|\frac{1}{Z}p_{\boldsymbol{\theta}}(\boldsymbol{y}_t|\boldsymbol{X}_t, \boldsymbol{z})q(\boldsymbol{z}|D_c)\right.\right). \tag{10}$$

If we identify the approximate posterior $q_{\boldsymbol{\phi}}$ with the encoder of the maximum-likelihood ConvNP, (which in the maximum-likelihood framework does not have an approximate inference interpretation), then $\log Z = \mathcal{L}_{\text{ML}}(\boldsymbol{\theta}, \boldsymbol{\phi}; \xi)$.

# G   Effect of Number of Samples Used to Estimate Objective During Training and Evaluation

In this section we empirically examine the effect of $L$, the number of samples used to estimate likelihood bounds, on the training and evaluation of ConvNPs and ANPs.

## G.1   Effect of Number of Samples Used for Evaluation

As the true log-likelihoods of NP-based models are intractable, quantitative evaluation and comparison of models is challenging. Instead, we compare models by using an estimate of the log-likelihood. A natural candidate is $\mathcal{L}_{\text{ML}}$. However, unless large $L$ is used, $\mathcal{L}_{\text{ML}}$ is conservative and tends to significantly underestimate the log-likelihood. One way to improve the estimate of $\mathcal{L}_{\text{ML}}$ is through importance weighting (IW) [18, 11]. Denoting $D = D_c \cup D_t$, the encoder $\text{E}_{\boldsymbol{\phi}}(D)$ can be used as a proposal distribution:

$$\hat{\mathcal{L}}_{\text{IW}}(\boldsymbol{\theta}, \boldsymbol{\phi}; \xi) := \log\left(\frac{1}{L}\sum_{l=1}^{L}\exp\left(\log w(\boldsymbol{z}_l) + \sum_{(\boldsymbol{x}, y) \in D_t}\log p_{\boldsymbol{\theta}}(y|\boldsymbol{x}, \boldsymbol{z}_l)\right)\right), \quad \boldsymbol{z}_l \sim \text{E}_{\boldsymbol{\phi}}(D),$$

$$\tag{11}$$

where the importance weights are given by $\log w(\boldsymbol{z}_l) := \log q_{\boldsymbol{\phi}}(\boldsymbol{z}|D_c) - \log q_{\boldsymbol{\phi}}(\boldsymbol{z}|D)$. Here $q_{\boldsymbol{\phi}}(\boldsymbol{z}|D)$ is the density of the encoder distribution. We find that training models with $\mathcal{L}_{\text{ML}}$ results in encoders that are ill-suited as proposal distributions, so we only use $\mathcal{L}_{\text{IW}}$ to evaluate models trained with $\mathcal{L}_{\text{NP}}$.

(a) Matérn$-\frac{5}{2}$

(b) Weakly periodic kernel

Figure 1: Log-likelihood bounds achieved by various combination of models and training objectives when evaluated with $\mathcal{L}_{\mathrm{ML}}$ and $\mathcal{L}_{\mathrm{IW}}$ for various numbers of samples $L$. Color indicates model. Solid lines correspond to models trained and evaluated with $\mathcal{L}_{\mathrm{ML}}$. Dashed lines correspond to models trained with $\mathcal{L}_{\mathrm{NP}}$ and evaluated with $\mathcal{L}_{\mathrm{IW}}$. Dotted lines correspond to models trained with $\mathcal{L}_{\mathrm{ML}}$ and evaluated with $\mathcal{L}_{\mathrm{ML}}$.

Fig 1 demonstrates the effect of the number of samples $L$ used to estimate the evaluation objective for the ConvNP and ANP trained with $\mathcal{L}_{\mathrm{ML}}$ and $\mathcal{L}_{\mathrm{NP}}$. The models used to generate Fig 1 are the same models used in Sec 5.1, i.e. having heteroskedastic noise. Observe the general trend that the log-likelihood estimates tend to increase with $L$, as expected. The ANP trained with $\mathcal{L}_{\mathrm{NP}}$ collapsed to a conditional ANP, meaning that the encoder became deterministic; in that case, $\mathcal{L}_{\mathrm{ML}}$ is exact, which means that larger $L$ and importance weighting will not increase the estimate. In contrast, the ANP trained with $\mathcal{L}_{\mathrm{ML}}$ did not collapse, and we see that there the estimate increases with $L$. For the ConvNP trained with $\mathcal{L}_{\mathrm{NP}}$, evaluating with $\mathcal{L}_{\mathrm{IW}}$ yields a significant increase, showing that the bound estimated with $\mathcal{L}_{\mathrm{IW}}$ is very loose. The models trained with $\mathcal{L}_{\mathrm{ML}}$ tend to be the best performing, although the ConvNP trained with $\mathcal{L}_{\mathrm{NP}}$ is best for weakly periodic kernel and appears to still be increasing with $L$.

In both the main and the supplement, all log-likelihood lower bounds reported are computed with $\mathcal{L}_{\mathrm{ML}}$ if the model was trained using $\mathcal{L}_{\mathrm{ML}}$ and with $\mathcal{L}_{\mathrm{IW}}$ if the model was trained using $\mathcal{L}_{\mathrm{NP}}$.

### G.2 Effect of Number of Samples Used During Training

Fig 2 shows the effect of the number of samples $L$ in the training objectives on the performance of the ConvNP and ANP. Observe that the performance of $\mathcal{L}_{\mathrm{ML}}$ reliably increases with the number of samples $L$ and that $\mathcal{L}_{\mathrm{ML}}$ outperforms $\mathcal{L}_{\mathrm{NP}}$. The performance for $\mathcal{L}_{\mathrm{NP}}$ does not appear to increase with the number of samples $L$ and appears more noisy than $\mathcal{L}_{\mathrm{ML}}$. Note that the models used for Fig 2

Figure 2: Interpolation performance (within training range) for context set sizes uniformly sampled from $\{0, \ldots, 50\}$ of the ConvNP and ANP on Matérn–$\frac{5}{2}$ samples. The models are trained with $\mathcal{L}_{\mathrm{ML}}$ and $\mathcal{L}_{\mathrm{NP}}$ for various number of samples $L$. Models trained with $\mathcal{L}_{\mathrm{ML}}$ are evaluated with $\mathcal{L}_{\mathrm{ML}}$, while models trained with $\mathcal{L}_{\mathrm{NP}}$ are evaluated with $\mathcal{L}_{\mathrm{ML}}$. At evaluation, all bounds are estimated using 2,048 samples.

were trained with homoskedastic observation noise. This is achieved by pooling $f_\sigma$ over the time dimension.

## H  Experimental Details on 1D Regression

For the full results of the 1D regression tasks, see App I. Code to reproduce the 1D regression experiments can be found at `https://github.com/wesselb/NeuralProcesses.jl`.

In the 1D regression experiments, we consider the following generative processes:

EQ: samples from a Gaussian process with the following exponentiated-quadratic kernel:

$$k(t, t') = \exp\left(-\frac{1}{8}(t - t')^2\right);$$

Matérn–$\frac{5}{2}$: samples from a Gaussian process with the following Matérn–$\frac{5}{2}$ kernel:

$$k(t, t') = \left(1 + 4\sqrt{5}d + \frac{5}{3}d^2\right)\exp\left(-\sqrt{5}d\right)$$

with $d = 4|x - x'|$;

noisy mixture: samples from a Gaussian process with the following noisy mixture kernel:

$$k(t, t') = \exp\left(-\frac{1}{8}(t - t')^2\right) + \exp\left(-\frac{1}{2}(t - t')^2\right) + 10^{-3}\delta[t - t'];$$

weakly periodic: samples from a Gaussian process with the following weakly-periodic kernel:

$$k(t, t') = \exp\left(-\frac{1}{2}(f_1(t) - f_1(t'))^2 - \frac{1}{2}(f_2(t) - f_2(t'))^2 - \frac{1}{8}(t - t')^2\right)$$

with $f_1(t) = \cos(8\pi t)$ and $f_2(t) = \sin(8\pi t)$; and

sawtooth: samples from the following sawtooth process:

$$f(t) = \frac{A}{2} - \frac{A}{\pi}\sum_{k=1}^{K}(-1)^k \frac{\sin(2\pi k f(t - s))}{k}$$

with $A = 1$, $f \sim \mathcal{U}[3,5]$, $s \sim \mathcal{U}[-5,5]$, and $K \in \{10, \ldots, 20\}$ chosen uniformly.

We compare the following models, where all activation functions are leaky ReLUs with leak $0.1$:

ConvCNP: The first model is the ConvCNP. The architecture of the ConvCNP is equal to that of the encoder in the ConvNP, described next.

ConvNP: The second model is the ConvNP as described in the main body. The functional embedding uses separate length scales for the data channel and density channel (**??**), which are initialized to twice the inter-point spacing of the discretization and learned during training. The discretization uniformly ranges over $[\min(x) - 1, \max(x) + 1]$ at density $\rho = 64$ points per unit, where $\min(x)$ is the minimum $x$ value occurring in the union of the context and target sets in the current batch and $\max(x)$ is corresponding maximum $x$ value. The discretization is passed through a 10-layer (excluding an initial and final point-wise linear layer) CNN with $64$ channels and depthwise-separable convolutions. The width of the filters depends on the data set and is chosen such that the receptive field sizes are as follows:

$$
\begin{aligned}
\text{EQ:} &\quad 2, \\
\text{Matérn-}\tfrac{5}{2}\text{:} &\quad 2, \\
\text{noisy mixture:} &\quad 4, \\
\text{weakly periodic:} &\quad 4, \\
\text{sawtooth:} &\quad 16.
\end{aligned}
$$

The discretized functional representation consists of 16 channels. The smoothing at the end of the encoder also has separate length scales for the mean and variance which are initialized similarly and learned. The encoder parametrizes the standard deviations by passing the output of the CNN through a softplus. The decoder has the same architecture as the encoder.

ANP: The third model is the Attentive NP with latent dimensionality $d = 128$ and 8-head dot-product attention [17]. In the attentive deterministic encoder, the keys ($t$), queries ($t$), and values (concatenation of $t$ and $y$) are transformed by a three-layer MLP of constant width $d$. The dot products are normalised by $\sqrt{d}$. The output of the attention mechanism is passed through a constant-width linear layer, which is then passed through two layers of layer normalization [1] to normalise the latent representation. In the first of these two layers, first the transformed queries are passed through a constant-width linear layer and added to the input. In the second of these two layers, the output of the first layer is first passed through a two-layer constant-width MLP and added to itself, making a residual layer. In the stochastic encoder, the inputs and outputs are concatenated and passed though a three-layer MLP of constant width $d$. The result is mean-pooled and passed through a two-layer constant-width MLP. The decoder consists of a three-layer MLP of constant width $d$.

NP: The fourth model is the original NP [4]. The architecture is similar to that of the ANP, where the architecture of the deterministic encoder is replaced by that of the stochastic encoder.

For all models, positivity of the observation noise is enforced with a softplus function. Parameter counts of the ConvCNP, ConvNP, ANP, and NP are listed in Tab 1.

The models are trained with $\mathcal{L}_{\mathrm{ML}}$ ($L = 20$) and $\mathcal{L}_{\mathrm{NP}}$ ($L = 5$). For $\mathcal{L}_{\mathrm{NP}}$, the context set is appended to the target set when evaluating the objective. The models are optimised using ADAM with learning rate $5 \cdot 10^{-3}$ for 100 epochs. One epoch consists of $2^{14}$ tasks divided into batches of size 16. For training, the inputs of the context and target sets are sampled uniformly from $[-2, 2]$. The size of the context set is sampled uniformly from $\{0, \ldots, 50\}$ and the size of the target set is fixed to 50. To encourage the NP-based models—not the CNP-based models—to fit and not revert to their conditional variants, the observation noise standard deviation $\sigma$ is held fixed to $10^{-2}$ for the first 20 epochs.

|         | EQ      | Matérn$-\frac{5}{2}$ | Noisy Mixt. | Weakly Per. | Sawtooth |
|---------|---------|----------------------|-------------|-------------|----------|
| ConvCNP | 42 822  | 42 822               | 51 014      | 51 014      | 100 166  |
| ConvNP  | 88 486  | 88 486               | 104 870     | 104 870     | 203 174  |
| ANP     | 530 178 | 530 178              | 530 178     | 530 178     | 530 178  |
| NP      | 479 874 | 479 874              | 479 874     | 479 874     | 479 874  |

Table 1: Parameter counts for the ConvCNP, ConvNP, ANP, and NP in the 1D regression tasks

For evaluation, the size of the context set is sampled uniformly from $\{0, \ldots, 10\}$, and the losses are evaluated with $L = 5000$ and batch size one. To test interpolation within the training range, the inputs of the context and target sets are, like training, sampled uniformly from $[-2, 2]$. To test interpolation beyond the training range, the inputs of the context and target sets are sampled uniformly from $[2, 6]$. To test extrapolation beyond the training range, the inputs of the context sets are sampled uniformly from $[-2, 2]$ and the inputs of the target sets are sampled uniformly from $[-4, -2] \cup [2, 4]$. As described in App G.1, models trained with $\mathcal{L}_{\mathrm{NP}}$ are evaluated using importance weighting to obtain a better estimate of the evaluation loss.

## I  Additional Results on 1D Regression

Tab 2 presents results for all models with all losses on all data sets described in App H according to the evaluation protocol described in Apps G.1 and H.

## J  Experimental Details on Image Completion

### J.1  Data Details

(a) Train ($32 \times 32$)          (b) Test ($56 \times 56$)

Figure 3: Samples from our generated Zero Shot Multi MNIST (ZSMM) data set.

We use three standard data sets throughout our image experiments: SVHN [13], MNIST [12], and $32 \times 32$ CelebA [13]. The aforementioned standard data sets all contain only a single, well-centered object. To evaluate the translation equivariance and generalization capabilities of our model we evaluate on a Zero Shot Multi-MNIST (ZSMM) task, which is similar to ZSMM described in Appendix D.2 of [5]. Namely, we generate a test set by randomly sampling with replacement 10000 pairs of digits from the MNIST test set, place them on a black $56 \times 56$ background, and translate the digits in such a way that the digits can be arbitrarily close but cannot overlap (Fig 3b). The difference with the dataset from Gordon et al. [5], is that the training set consists of the standard MNIST digits (instead of a single digit placed in the center of $56 \times 56$ canvas), augmented by up to 4 pixel shifts (Fig 3a). The model thus has to generalize both to a larger canvas size as well as to seeing multiple digits.

Table 2: Log-likelihood for ConvCNP, ConvNP, ANP, and NP. Each of the stochastic models was trained on each data set with $\mathcal{L}_{\mathrm{ML}}$ and $\mathcal{L}_{\mathrm{NP}}$, separately.

| | | EQ | Matérn–$\frac{5}{2}$ | Noisy Mixt. | Weakly Per. | Sawtooth |
|---|---|---|---|---|---|---|
| INTERPOLATION INSIDE TRAINING RANGE | | | | | | |
| GP (full) | | $5.80 \pm 0.02$ | $1.22 \pm 6.3\text{E}{-}3$ | $1.00 \pm 4.1\text{E}{-}3$ | $-0.06 \pm 4.6\text{E}{-}3$ | N/A |
| GP (diag) | | $-0.59 \pm 0.01$ | $-0.84 \pm 9.0\text{E}{-}3$ | $-0.89 \pm 0.01$ | $-1.17 \pm 5.2\text{E}{-}3$ | N/A |
| ConvCNP | | $-0.70 \pm 0.02$ | $-0.88 \pm 0.01$ | $-0.92 \pm 0.02$ | $-1.19 \pm 7.0\text{E}{-}3$ | $1.15 \pm 0.04$ |
| ConvNP | $\mathcal{L}_{\mathrm{ML}}$ | $-0.30 \pm 0.02$ | $-0.58 \pm 0.01$ | $-0.55 \pm 0.01$ | $-1.02 \pm 6.0\text{E}{-}3$ | $2.30 \pm 0.01$ |
| ANP | $\mathcal{L}_{\mathrm{ML}}$ | $-0.52 \pm 0.01$ | $-0.73 \pm 0.01$ | $-0.69 \pm 0.01$ | $-1.14 \pm 6.0\text{E}{-}3$ | $0.09 \pm 3.0\text{E}{-}3$ |
| NP | $\mathcal{L}_{\mathrm{ML}}$ | $-0.84 \pm 9.0\text{E}{-}3$ | $-0.96 \pm 7.0\text{E}{-}3$ | $-0.93 \pm 9.0\text{E}{-}3$ | $-1.23 \pm 5.0\text{E}{-}3$ | $-0.02 \pm 2.0\text{E}{-}3$ |
| ConvNP | $\mathcal{L}_{\mathrm{NP}}$ | $-0.50 \pm 0.02$ | $-0.77 \pm 0.01$ | $-0.48 \pm 0.02$ | $-1.03 \pm 8.0\text{E}{-}3$ | $2.47 \pm 8.0\text{E}{-}3$ |
| ANP | $\mathcal{L}_{\mathrm{NP}}$ | $-0.82 \pm 0.01$ | $-0.96 \pm 0.01$ | $-1.04 \pm 0.01$ | $-1.37 \pm 6.0\text{E}{-}3$ | $0.20 \pm 9.0\text{E}{-}3$ |
| NP | $\mathcal{L}_{\mathrm{NP}}$ | $-0.58 \pm 9.0\text{E}{-}3$ | $-1.00 \pm 9.0\text{E}{-}3$ | $-0.72 \pm 0.01$ | $-1.22 \pm 5.0\text{E}{-}3$ | $-0.16 \pm 2.0\text{E}{-}3$ |
| INTERPOLATION BEYOND TRAINING RANGE | | | | | | |
| GP (full) | | $5.80 \pm 0.02$ | $1.22 \pm 6.3\text{E}{-}3$ | $1.00 \pm 4.1\text{E}{-}3$ | $-0.06 \pm 4.6\text{E}{-}3$ | N/A |
| GP (diag) | | $-0.59 \pm 0.01$ | $-0.84 \pm 9.0\text{E}{-}3$ | $-0.89 \pm 0.01$ | $-1.17 \pm 5.2\text{E}{-}3$ | N/A |
| ConvCNP | | $-0.69 \pm 0.02$ | $-0.87 \pm 0.01$ | $-0.94 \pm 0.02$ | $-1.19 \pm 7.0\text{E}{-}3$ | $1.11 \pm 0.04$ |
| ConvNP | $\mathcal{L}_{\mathrm{ML}}$ | $-0.30 \pm 0.02$ | $-0.58 \pm 0.01$ | $-0.56 \pm 0.01$ | $-1.03 \pm 6.0\text{E}{-}3$ | $2.29 \pm 0.02$ |
| ANP | $\mathcal{L}_{\mathrm{ML}}$ | $-1.35 \pm 6.0\text{E}{-}3$ | $-1.39 \pm 7.0\text{E}{-}3$ | $-1.65 \pm 5.0\text{E}{-}3$ | $-1.35 \pm 4.0\text{E}{-}3$ | $-0.17 \pm 1.0\text{E}{-}3$ |
| NP | $\mathcal{L}_{\mathrm{ML}}$ | $-2.70 \pm 3.0\text{E}{-}3$ | $-2.60 \pm 3.0\text{E}{-}3$ | $-2.82 \pm 3.0\text{E}{-}3$ | - | $-0.03 \pm 2.0\text{E}{-}3$ |
| ConvNP | $\mathcal{L}_{\mathrm{NP}}$ | $-0.48 \pm 0.02$ | $-0.79 \pm 0.01$ | $-0.48 \pm 0.02$ | $-1.04 \pm 8.0\text{E}{-}3$ | $2.47 \pm 8.0\text{E}{-}3$ |
| ANP | $\mathcal{L}_{\mathrm{NP}}$ | $-1.91 \pm 0.03$ | $-1.48 \pm 4.0\text{E}{-}3$ | $-1.85 \pm 7.0\text{E}{-}3$ | $-1.66 \pm 0.01$ | $-0.30 \pm 4.0\text{E}{-}3$ |
| NP | $\mathcal{L}_{\mathrm{NP}}$ | $-13.7 \pm 0.82$ | $-3.96 \pm 0.04$ | $-3.80 \pm 0.02$ | - | $-4.98 \pm 0.02$ |
| EXTRAPOLATION BEYOND TRAINING RANGE | | | | | | |
| GP (full) | | $4.29 \pm 6.2\text{E}{-}3$ | $0.82 \pm 4.3\text{E}{-}3$ | $0.66 \pm 2.2\text{E}{-}3$ | $-0.33 \pm 3.4\text{E}{-}3$ | N/A |
| GP (diag) | | $-1.40 \pm 5.0\text{E}{-}3$ | $-1.41 \pm 4.8\text{E}{-}3$ | $-1.72 \pm 6.2\text{E}{-}3$ | $-1.40 \pm 4.0\text{E}{-}3$ | N/A |
| ConvCNP | | $-1.41 \pm 6.0\text{E}{-}3$ | $-1.41 \pm 7.0\text{E}{-}3$ | $-1.73 \pm 8.0\text{E}{-}3$ | $-1.41 \pm 6.0\text{E}{-}3$ | $0.27 \pm 0.02$ |
| ConvNP | $\mathcal{L}_{\mathrm{ML}}$ | $-1.09 \pm 5.0\text{E}{-}3$ | $-1.11 \pm 5.0\text{E}{-}3$ | $-1.30 \pm 4.0\text{E}{-}3$ | $-1.24 \pm 4.0\text{E}{-}3$ | $1.61 \pm 0.02$ |
| ANP | $\mathcal{L}_{\mathrm{ML}}$ | $-1.29 \pm 6.0\text{E}{-}3$ | $-1.29 \pm 5.0\text{E}{-}3$ | $-1.55 \pm 5.0\text{E}{-}3$ | $-1.34 \pm 5.0\text{E}{-}3$ | $-0.25 \pm 2.0\text{E}{-}3$ |
| NP | $\mathcal{L}_{\mathrm{ML}}$ | $-2.23 \pm 4.0\text{E}{-}3$ | $-2.08 \pm 3.0\text{E}{-}3$ | $-2.50 \pm 4.0\text{E}{-}3$ | $-1.39 \pm 4.0\text{E}{-}3$ | $-0.06 \pm 2.0\text{E}{-}3$ |
| ConvNP | $\mathcal{L}_{\mathrm{NP}}$ | $-1.21 \pm 0.01$ | $-1.31 \pm 0.01$ | $-1.19 \pm 0.01$ | $-1.51 \pm 8.0\text{E}{-}3$ | $2.10 \pm 7.0\text{E}{-}3$ |
| ANP | $\mathcal{L}_{\mathrm{NP}}$ | $-1.44 \pm 6.0\text{E}{-}3$ | $-1.45 \pm 6.0\text{E}{-}3$ | $-1.77 \pm 7.0\text{E}{-}3$ | $-1.46 \pm 6.0\text{E}{-}3$ | $-0.20 \pm 2.0\text{E}{-}3$ |
| NP | $\mathcal{L}_{\mathrm{NP}}$ | $-5.85 \pm 0.05$ | $-2.65 \pm 3.0\text{E}{-}3$ | $-4.06 \pm 0.04$ | $-1.49 \pm 5.0\text{E}{-}3$ | $-1.99 \pm 6.0\text{E}{-}3$ |

For all data sets, pixel values are divided by 255 to rescale them to the $[0, 1]$ range. We evaluate on predefined test splits when available (MNIST, SVHN, ZSMM) and make our own test set for CelebA by randomly selecting $10\%$ of the data. For each dataset we also set aside $10\%$ of the training set as validation.

## J.2 Training Details

In all experiments, we sample the number of context pixels uniformly from $\mathcal{U}(0, \frac{n_{\mathrm{total}}}{2})$, and the number of target points is set to $n_{\mathrm{total}}$. The weights are optimized using Adam [9] with learning rate $5 \times 10^{-4}$. We use a maximum of 100 epochs, with early stopping — based on log likelihood on the validation set — of 10 epochs patience. Unless stated otherwise, we use $L = 16$ samples from the latent function during training, and $L = 128$ at test time. We clip the $L2$ norm of all gradients to 1, which was particularly important for ConvNP. We use a batch size of 32 for all models besides ANP trained on ZSMM which used a batch size of 8 due to memory constraints.

## J.3 Architecture Details

**General architecture details.** For all models, we follow Le et al. [11] and process the predicted standard deviation of the latent function $\boldsymbol{\sigma}_z$ using a sigmoid and the standard deviation $\boldsymbol{\sigma}$ of the

predictive distribution using lower-bounded softplus:

$$\boldsymbol{\sigma}_z = 0.001 + (1 - 0.001)\frac{1}{1 + \exp(f_{\sigma,z})} \tag{12}$$

$$\boldsymbol{\sigma} = 0.001 + (1 - 0.001)\ln(1 + \exp(f_\sigma)) \tag{13}$$

As the pixels are rescaled to $[0, 1]$, we also process the mean of the posterior predictive (conditioned on a single sample) to be in $[0, 1]$ using a logistic function

$$\boldsymbol{\mu} = \frac{1}{1 + \exp(-f_\mu)} \tag{14}$$

In the following, we describe the architecture of ANP and ConvNP. Unless stated otherwise, all vectors in the following paragraphs are in $\mathbb{R}^{128}$ and all MLPs have 128 hidden units.

**ANP details.** We provide details for the ANP trained with $\mathcal{L}_{\mathrm{ML}}$. As the ANP cannot take advantage of the fact that images are on the grid, we preprocess each pixel so that $\mathbf{x} \in [-1, 1]^2$. The only exception being for the test set of ZSMM, where $\mathbf{x} \in [-\frac{56}{32}, \frac{56}{32}]^2$ as the model is trained on $32 \times 32$ but evaluated on $56 \times 56$ images. Each context feature is first encoded $\mathbf{x}^{(c)} \mapsto \mathbf{r}_x^{(c)}$ by a single hidden layer MLP, while a second single hidden layer MLP encodes values $\mathbf{y}^{(c)} \mapsto \mathbf{r}_y^{(c)}$. We produce a representation $\mathbf{r}_{xy}^{(c)}$ by summing both representations $\mathbf{r}_x^{(c)} + \mathbf{r}_y^{(c)}$ and passing them through two self-attention layers [17]. Following Parmar et al. [14], each self-attention layer is implemented as 8-headed attention, a skip connection, and two layer normalizations [1]. To predict values at each target point $t$, we embed $\mathbf{x}^{(t)} \mapsto \mathbf{r}_x^{(t)}$ using the hidden layer MLP used for $\mathbf{r}_x^{(c)}$. A deterministic target representation $\mathbf{r}_{xy}^{(t)}$ is then computed by applying cross-attention (using an 8-headed attention described above) with keys $\mathrm{K} := \{\mathbf{r}_x^{(c)}\}_{c=1}^C$, values $\mathrm{V} := \{\mathbf{r}_{xy}^{(c)}\}_{c=1}^C$, and query $\mathbf{q} := \mathbf{r}_x^{(t)}$. For the latent path, we average over context representations $\mathbf{r}_{xy}^{(c)}$, and pass the resulting representation through a single hidden layer MLP that outputs $(\boldsymbol{\mu}_z, \boldsymbol{\sigma}_z) \in \mathbb{R}^{256}$. $\boldsymbol{\sigma}_z$ is made positive by post-processing it using Eq (12). We then sample (with reparametrization [10]) $L$ latent representation $\mathbf{z}_l \sim \mathcal{N}(\mathbf{z}; \boldsymbol{\mu}_z, \boldsymbol{\sigma}_z^2)$.

We describe the remainder of the forward pass for a single $\mathbf{z}_l$, though in practice multiple samples may be processed in parallel. The deterministic and latent representations of the context set are concatenated, and the resulting representation is passed through a linear layer $[\mathbf{r}_{xy}^{(t)}; \mathbf{z}_l] \rightarrow \mathbf{r}_{xyz}^{(t)} \in \mathbb{R}^{128}$. Given the target and context-set representations, the predictive posterior is given by a Gaussian pdf with diagonal covariance parametrised by $(\boldsymbol{\mu}^{(t)}, \boldsymbol{\sigma}_{\mathrm{pre}}^{(t)}) = \mathrm{decoder}([\mathbf{r}_x^{(t)}; \mathbf{r}_{xyz}^{(t)}])$ where $\boldsymbol{\mu}^{(t)}, \boldsymbol{\sigma}_{\mathrm{pre}}^{(t)} \in \mathbb{R}^3$ and $\mathrm{decoder}$ is a 4 hidden layer MLP. Finally, the $\boldsymbol{\sigma}^{(t)}$ is processed by Eq (13) using Eq (14). In the case of MNIST and ZSMM, $\boldsymbol{\sigma}^{(t)}$ is also spatially mean pooled, which corresponds to using homoskedastic noise. This improves the qualitative performance by forcing ANP and ConvNP to model the digit instead of focusing on predicting the black background with high confidence. Kim et al. [8] did not suffer from that issue as they used a much larger lower bound for Eq (13).

**ConvNP details.** The core algorithm of on-the-grid ConvNP is outlined in Algorithm 4 as well as Algorithm 2. Here we discuss the parametrizations used for each step of the algorithm. All convolutional layers are depthwise separable [2]. $\mathrm{CONV}_\theta$ is a convolutional layer with kernel size of 11 (no bias). Following Gordon et al. [5], we enforce positivity on the weights in the first convolutional layer by only convolving their absolute value with the signal.

The CNNs are ResNets [6] with 9 blocks, where each convolution has a kernel size of 3. Each residual block consists of two convolutional layers, pre-activation batch normalization layers [7], and ReLU activations. The output of the pre-latent CNN (CNN in Algorithm 2) goes through a single hidden layer MLP that outputs $(\boldsymbol{\mu}_z, \boldsymbol{\sigma}_z) \in \mathbb{R}^{256}$. As with ANP, $f_{\sigma,z}$ is processed by Eq (12) and then used to sample (with reparametrization [10]) $L$ latent functions $\mathbf{Z}_l$. Importantly, we found that the coherence of samples improves if the model uses a *global representation* in addition to the the pixel dependent representation. We achieve this by mean-pooling half of the functional representation. Namely, we replace $\mathbf{z}_l$ by the channel-wise concatenation of $\mathbf{z}_l^{(1:64)}$ and $\mathrm{MEAN}(\mathbf{z}_l^{(65:128)})$, where the mean is taken over the spatial dimensions. This latent function then goes through the post-latent CNN (CNN in Algorithm 4), as well as a linear layer to output $(f_\mu, f_\sigma) \in \mathbb{R}^{256}$. As for ANP $f_\mu$ is processed by Eq (14) and $f_\sigma$ is re-scaled with Eq (13) and is spatially pooled in the case of MNIST and ZSMM to obtain homoskedastic noise.

# K  Additional results on image completion.

We provide additional qualitative samples and quantitative analyses for the ConvNP and ANP.

**Additional ConvNP samples.** Fig 4 provides further samples from a ConvNP trained with $\mathcal{L}_{\mathrm{ML}}$. We observe that the ConvNP produces reasonably diverse yet coherent samples when evaluated in a regime that resembles the training regime (in the first four sub-columns of MNIST, SVHN, and CelebA). However, Fig 4 also demonstrates that the ConvNP struggles with context sets that are significantly different from those seen during training.

**Further comparisons of ANP and ConvNP.** We provide further qualitative comparisons of Con­vNPs, ANPs trained with $\mathcal{L}_{\mathrm{ML}}$, and ANPs trained with $\mathcal{L}_{\mathrm{NP}}$. We omit ConvNPs trained with $\mathcal{L}_{\mathrm{NP}}$ as these are significantly outperformed by ConvNPs trained with $\mathcal{L}_{\mathrm{ML}}$ (see e.g. Tab 2).

Fig 5 shows that all models perform relatively well when context sets are drawn from a similar distribution as employed during training (first four sub-columns of MNIST, SVHN, and CelebA). Furthermore, we observe that samples from the ConvNP prior tend to be closer to samples from the underlying data distribution (e.g. for CelebA).

The qualitative advantage of ConvNP is most significant in settings that require translation equivari­ance for generalization. Fig 5 row 2 (ZSMM) clearly demonstrates that ConvNP generalizes to larger canvas sizes and multiple digits, while ANP attempts to reconstruct a single digit regardless of the context set. Finally, Fig 6 provides the test log-likelihood distributions of ANP and ConvNP as well as some qualitative comparisons between the two.

# L  Experimental Details on Environmental Data

## L.1  Data Details

Table 3: Coordinates for boxes defining the train and test regions. Latitidues are given as (north, south), and longitudes as (west, east).

|            | Central (train) | Western (test) | Eastern (test) | Southern (test) |
|------------|-----------------|----------------|----------------|-----------------|
| Latitudes  | $(52, 46)$      | $(50, 46)$     | $(52, 49)$     | $(46, 42)$      |
| Longitudes | $(08, 28)$      | $(01, 08)$     | $(28, 35)$     | $(19, 26)$      |

ERA5-Land [15] contains high resolution information on environmental variables at a 9 km spacing across the globe.[6] The data we use contains daily measurements of accumulated precipitation at 11pm and temperature at 11pm at every location, between 1981 and 2020, yielding a total of 14,304 temporal measurements across the spatial grid. In addition, we provide orography (elevation) values for each location. We normalize the data such that the precipitation values in the train set have zero mean and unit standard deviation.

We consider the task of predicting daily precipitation $y$, with latitude and longitude as $\boldsymbol{x}$. In addition, at each context and target location, we provide the model with access to side information in the form of orography (elevation) and temperature values. We also normalize the orography and temperature values to have zero mean and unit standard deviation. We choose a large region of central Europe as our train set, and use regions East, West and South of the train set as held out test sets (see Fig 7 and Tab 3). At train time, to sample a task, we first sample a random date between 1981 and 2020. We then sample a square subregion of grid of values from within the train region (which has size $61 \times 201$). We consider two models, one trained on $28 \times 28$ subregions, and another trained on $40 \times 40$ subregions. During training, each subregion is then split into context and target sets. Context points are randomly chosen with a keep rate $p_{\mathrm{keep}}$ with $p_{\mathrm{keep}} \sim \mathcal{U}[0, 0.3]$. In this section, we train only on the $\mathcal{L}_{\mathrm{ML}}$ objective.

## L.2 Gaussian Process Baseline

We mean-centre the data for each task for the GP before training, and add the mean offset back for evaluation and sampling. We use an Automatic Relevance Determination (ARD) kernel, with separate factors for latitude/longitude, temperature and orography. In detail, let $\boldsymbol{x} = (x_{\text{lat}}, x_{\text{lon}})$ denote position, and let $\omega, t$ denote orography and precipitation respectively, and let $\boldsymbol{r} \coloneqq (\boldsymbol{x}, \omega, t)$. Then the kernel is given by

$$k(\boldsymbol{r}, \boldsymbol{r}') = \sigma_v^2 k_l(\boldsymbol{x}, \boldsymbol{x}') k_\omega(\omega, \omega') k_t(t, t') + \sigma_n^2 \delta(\boldsymbol{r}, \boldsymbol{r}').$$

Here each of $k_l, k_\omega$ and $k_t$ are Matérn–$\frac{5}{2}$ kernels with separate learnable lengthscales; $\delta(\boldsymbol{r}, \boldsymbol{r}') = 1$ if $\boldsymbol{r} = \boldsymbol{r}'$ and 0 otherwise; and $\sigma_v^2, \sigma_n^2$ are learnable signal and noise variances respectively. We learn all hyperparameters by maximising the log-marginal likelihood using Scipy's implementation of L-BFGS.

**Transforming the data.** As the data is non-negative, we considered applying the transform $y \mapsto \log(\epsilon + y)$ for the GP to model. If $\epsilon = 0$, this would guarantee that the GP would only yield positive samples, which would be physically sensible as precipitation is non-negative. However, this cannot be done as precipitation often takes the value $y = 0$, which would lead to the transform being undefined. On the other hand, if $\epsilon > 0$, the GP samples after performing the inverse transform could still predict a precipitation value as low as $-\epsilon$, which is still unphysical. Further, a small value of $\epsilon$ leads to large distortion of the $y$ values in transformed space. In the end, we run all experiments for the GP and NP without log-transforming the data; hence the models have to learn non-negativity.

## L.3 ConvNP Architecture and Training Details

As the ERA5-Land dataset is regularly spaced, we use the on-the-grid version of the architecture, without the need for an RBF smoothing layer at the input (see App C). All experiments used a convolutional architecture with 3 residual blocks [6] for the encoder and 3 residual blocks for the decoder. Each residual block is defined with two layers of ReLU activations followed by convolutions, each with kernel size 5. The first convolution in each block is a standard convolution layer, whereas the second is depthwise separable [2]. All intermediate convolutional layers have 128 channels, and the latent function $\boldsymbol{z}$ has 16 channels. The networks were trained using ADAM with a learning rate of $10^{-4}$. We used 16 channels for the latent function $\boldsymbol{z}$, and estimated $\mathcal{L}_{\text{ML}}$ using 16-32 samples at train time, with batches of 8-16 images.

We train the models for between 400 and 500 epochs, where each epoch is defined as a single pass through each day in the training set, where at each day, a random subregion of the full $61 \times 201$ central Europe region is cropped. We estimated the predictive density using 2500 samples of $\boldsymbol{z}$ during test time.

## L.4 Prediction and Sampling

To create Tab 3, at test time we sample $28 \times 28$ subregions from each of the train and test regions. This is done 1000 times. For the GP, we randomly restart optimisation 5 times per task and use the best hyper-parameters found. In order to remove outliers where the GP has very poor likelihood, we set a log-likelihood threshold for the GP. If the GP has a log-likelihood of less than 0 nats on a particular task, then that task is removed from the evaluation.

We find that to produce high quality samples, we need to train the model on subregions that are roughly as large as the lengthscale of the precipitation process. Hence we sample from the model trained on $40 \times 40$ subregions in Fig 5 in the main body. We show samples from the model trained on both $28 \times 28$ subregions and $40 \times 40$ subregions in App M. We also compare to samples from GPs trained on each context set (no random restarts were used for sampling).

## L.5 Bayesian Optimization

We use the models described in App L.3, trained on random $28 \times 28$ subregions of the train region, and compare to the GP baselines described in App L.2. For the Bayesian optimization experiments in Fig 6 in the main body, we do not perform random restarts as this was too time-consuming. We carry out the Bayesian optimization (BayesOpt) experiments in each of the four regions: Central (train), West (test), East (test), and South (test). Each Bayesian optimization "episode" is defined by randomly sub-sampling a day (uniformly at random between 1981 and 2020), then sampling a sub-region from the tested region. To test the models' spatial generalization capacity (where possible),

we sub-sample episodes from each of the four regions with the following sizes: (i) Central: 42x42, (ii) West: 40x40, (iii) East: 28x28, and (iv) South: 36x36.

Episodes begin from empty sets $D_c^{(0)} =$, and models sequentially query locations for $t = 1, \ldots, 50$. Denoting $(\boldsymbol{x}^{(t)}, y^{(t)})$ the query location and queried value at iteration $t$, the context set is then updated as $D_c^{(t)} = D_c^{(t-1)} \cup \{(\boldsymbol{x}^{(t)}, y^{(t)})\}$. Denoting $\boldsymbol{y}$ as the complete set of rainfall values in the sub-region, and $\boldsymbol{y}^{(t)}$ as the set of queried values at iteration $t$, we can define the *instantaneous regret* as $r_t = \max(\boldsymbol{y}) - \max(\boldsymbol{y}_c^{(t)})$, and compute the average regret (plotted in Fig 6 in the main text) at the $t^{\text{th}}$ iteration as $\bar{r}_t = \frac{1}{t} \sum_{i=1}^{t} r_i$.

## M Additional Figures for Environmental Data

### M.1 Predictive density

Fig 8 displays the predictive densities for precipitation at different locations, conditioned on a context set used for testing. The density of the ConvNP is estimated using 2500 samples of $\boldsymbol{z}$. To examine why the ConvNP outperforms the GP in terms of log-likelihood, we plot cases where the ConvNP likelihood is significantly better than the GP likelihood. We see that this is due to the GP occasionally making very overconfident predictions compared to the ConvNP. We also see that the ConvNP in a small proportion of cases exhibits very non-Gaussian, asymmetric predictive distribtuions.

### M.2 Additional Samples

In this section we show additional samples from the model trained on $28 \times 28$ images (Figs 9 and 10) and also on $40 \times 40$ images (Figs 11 and 12). Training on larger images reduces the occurence of blocky artefacts. Fig 5 in the main body was trained on $40 \times 40$ images. Note that samples shown here are $61 \times 201$, i.e. the size of the entire central Europe train region.

Figure 4: Qualitative samples for one of the ConvNP trained with $\mathcal{L}_{\mathrm{ML}}$ in Tab 2. From top to bottom the four major rows correspond to MNIST, ZSMM, SVHN, CelebA32 datasets. For each dataset and each of the two major columns, a different image is randomly sampled; the first sub-row shows the given context points (missing pixels are in blue for MNIST and ZSMM but in black for SVHN and CelebA), while the next three sub-rows show the mean of the posterior predictive corresponding to different samples of the latent function. To show diverse samples we select three samples that maximize the average Euclidean distance between pixels of the samples. From left to right the first four sub-columns correspond to a context set with 0%, 1%, 3%, 10% randomly sampled context points. In the last two sub-columns, the context sets respectively contain all the pixels in the left and top half of the image.

Figure 5: Qualitative samples between (a) ConvNP trained with $\mathcal{L}_{\mathrm{ML}}$; (b) ANP trained with $\mathcal{L}_{\mathrm{ML}}$; (c) ANP trained with $\mathcal{L}_{\mathrm{NP}}$. For each model the figure shows the same as Fig 4.

(a) MNIST

(b) CelebA32

(c) Zero Shot Multi-MNIST

(d) SVHN

Figure 6: Log-likelihood and qualitative samples comparing ConvNP and ANP trained with $\mathcal{L}_{\mathrm{ML}}$ on (a) MNIST; (b) CelebA; (c) ZSMM; (d) SVHN. For each sub-figure, the top row shows the log-likelihood distribution for both models. The images below correspond to the context points (top), followed by three samples form ConvNP (mean of the posterior predictive corresponding to different samples from the latent function), and three samples from ANP. Each column corresponds to a given percentile of the ConvNP test log likelihood (as shown by green arrows).

Figure 7: Training (blue) and test (red) regions in Europe, along with orography data from ERA5Land.

(a)

(b)

Figure 8: Predictive density at two target points, where the ConvNP significantly outperforms the GP. The orange and blue circles show the likelihood of the ground truth target value under the GP and ConvNP. Note that as the precipitation values are normalized to zero mean and unit standard deviation, $y_t = -0.53$ corresponds to no rain. In Fig 8a, we see the ConvNP sometimes produces predictions heavily centered on this value, showing it has learned the sparsity of precipitation values. In Fig 8b we see the ConvNP predictive distribution is sometimes asymmetric with a heavier positive tail, reflecting the non-negativity of precipitation.

(a) Ground truth data    (b) ConvNP sample 1    (c) ConvNP sample 2    (d) ConvNP sample 3

(e) Context set    (f) GP sample 1    (g) GP sample 2    (h) GP sample 3

Figure 9: Samples from the predictive processes overlaid on central Europe, for a model trained on random $28 \times 28$ subregions of the full $61 \times 201$ central Europe region. Note some blocky artefacts in the ConvNP samples due to training on small subregions. Here the GP has overfit to the orography data, with samples that resemble the orography rather than precipitation.

(a) Ground truth data     (b) ConvNP sample 1     (c) ConvNP sample 2     (d) ConvNP sample 3

(e) Context set     (f) GP sample 1     (g) GP sample 2     (h) GP sample 3

Figure 10: Samples from the predictive processes overlaid on central Europe, for a model trained on random $28 \times 28$ subregions of the full $61 \times 201$ central Europe region. Here the GP has learned a lengthscale that is too large.

(a) Ground truth data     (b) ConvNP sample 1     (c) ConvNP sample 2     (d) ConvNP sample 3

(e) Context set     (f) GP sample 1     (g) GP sample 2     (h) GP sample 3

Figure 11: Samples from the predictive processes overlaid on central Europe, for a model trained on random $40 \times 40$ subregions of the full $61 \times 201$ central Europe region. Here the GP has overfit to the orography data, with samples that resemble the orography rather than precipitation.

(a) Ground truth data     (b) ConvNP sample 1     (c) ConvNP sample 2     (d) ConvNP sample 3

(e) Context set     (f) GP sample 1     (g) GP sample 2     (h) GP sample 3

Figure 12: Samples from the predictive processes overlaid on central Europe, for a model trained on random $40 \times 40$ subregions of the full $61 \times 201$ central Europe region. The GP has again overfit to the orography data.

(a) Ground truth data     (b) ConvNP sample 1     (c) ConvNP sample 2     (d) ConvNP sample 3

(e) Context set     (f) GP sample 1     (g) GP sample 2     (h) GP sample 3

Figure 13: Samples from the predictive processes overlaid on central Europe, for a model trained on random $40 \times 40$ subregions of the full $61 \times 201$ central Europe region.

## Footnotes

[3]To prevent notational clutter, the same symbol, $T_{\boldsymbol{\tau}}$, will denote translations on multiple kinds of objects.

[4]This is well-defined since $\Sigma$ is closed under translations. Equivalently, we could define $T_{\boldsymbol{\tau}} P$ as the push-forward of $P$ under the the translation map on functions, $T_{\boldsymbol{\tau}} : \mathbb{R}^{\mathcal{X}} \to \mathbb{R}^{\mathcal{X}}$.

[5]We exclude conditioning on observations that have zero density, so that the prediction map is well defined.

[6]URL: https://www.ecmwf.int/en/era5-land. Neither the European Commission nor ECMWF is responsible for any use that may be made of the Copernicus Information or data it contains.