[Reviews · NeurIPS 2020]

Review 1

Summary and Contributions: This paper proposes a model called convNP which adds a global latent variable to the previously proposed Convolutional Conditional Neural Process (convCNP), allowing it to produce coherent samples rather than independent samples per data-point (similar to what Neural Process is relative to Conditional Neural Process). The authors propose to train the model by approximating the likelihood objective using importance sampling, rather than amortized variational inferernce, and compare between the two both for the proposed model and for a previously proposed variant of NP that uses attention. The experiments show that the proposed model and training method outperform the baselines on prediction of 1D functions, images and weather data.

Strengths: The paper extends the convCNP model in a way that makes it useful in cases where there is a global uncertainty that needs to be captured and coherent samples are required. The paper proposed a different training method than what the rest of the work on NPs is usually using. The results also suggest that the method is better not only for the proposed model but also for previously proposed models like ANP (although this is still not very convincing in this paper and would require more research). The paper is well written and the experimental section does a good job in conveying the performance of the model, and shows that it outperforms strong baselines for the tested tasks.

Weaknesses: Some parts of the explanations contain statements that are a bit vague and hard to follow. The image completion experiments fail to show multi-modal prediction. see below for more details.

Correctness: As far as I can tell everything is correct. See some minor comments below on some unclear issues.

Clarity: The paper is well written.

Relation to Prior Work: The paper proposes a relatively small extension to a previous model (ConvCNP) and as such is described mainly in relation to it. As it is part of the general Neural Process family of models it is also compared clearly to other members of the family (NP) and the experiments are compared to another strong baseline from the family (ANP). One thing that I think is missing is comparing the performance in experiments to ConvCNP.

Reproducibility: Yes

Additional Feedback: 1. The authors say that they use as an encoder a convCNP. Looking at the psudo-code in algorithm 1 in the appendix, it is unclear to me if the convCNP is actually run all the way and given some discretize grid as targets, or are the discretization at the level of t_i used? I would assume the latter but this is not stated in the text. If it's the former I don't understand why line 6 and 7 (in algorithm 1) are needed in the encoder. 2. Figure 1 is not very informative specially compared to figure 1 in the appendix which is much better. Same goes for the pseudo-code in the appendix. I think it would be good to move those to the main paper even at the expense of some of the text. 3. Since both the proposed training method and the ELBO method are surrogate to maximum likelihood optimization, I would call the proposed method Importance Sampling (IS) and the ELBO method VI or ELBO. 4. I don't understand the claim in lines 172-173. Why is this not a lower bound? How is this different conceptually from the standard NP or any latent variable model? 5. Line 174-175 discuss the difficulty in using the KL divergence for 'non_discretized ConvNP' I don't see any reference to this model. As far as I understand convCNP always need some discretization to realize the method. 6. The image completion task in figure 3 don't show a prediction that covers different modes, which is an important aspect of NPs. I would add experiments with very few context points showing that the model can produce different figures, and if this doesn't happen that this should be discussed. Update: I have read the authors' feedback and happy to keep my score.


Review 2

Summary and Contributions: Updated: I have read the other reviews and comments and I will stick to my current score. The authors build on previous work on Convolutional Conditional Neural Processes (ConvCNPs) and introduce Convolutional Neural Processes (ConvNPs). Unlike ConvCNPs, ConvNPs can capture complex joint distributions between the predictions. In addition to ConvCNP they present a new training procedure based on approximate maximum likelihood (as opposed to variational inference, which was used for training previous latent variable Neural Process (NP) models). Finally, the authors provide experimental results on time-series few-shot regression experiments, image inpainting experiments as well as an environmental dataset.

Strengths: - The model is a natural extension of the original ConvCNP. The fact that ConcNPs can carry out few-shot regression on data with translation equivariances is a significant contribution. The additional maximum likelihood training regime makes this an even stronger paper. - Overall the paper is well written and clearly structured and has good figures. - The experiments are varied and the results look convincing. Normally I would not consider a single baseline to be enough, but in this case attentive NPs are really the main model that the authors should be comparing to (and GPs for tasks that require translation equivariance, as they do), so one baseline seems sufficient.

Weaknesses: The paper is dense in some parts, as it is building on previous work and lots of notation.

Correctness: The methodology seems correct.

Clarity: Yes.

Relation to Prior Work: Yes.

Reproducibility: Yes

Additional Feedback: No additional comments.


Review 3

Summary and Contributions: This paper proposes Convolutional Neural Process (ConvNP), which extends ConvCNP to model more expressive distribution especially for spatial temporal setting. Also, they proposes variational lower bound approach and aproximate maximum-likelihood training method for learning ConvNP. Experimental results on toy time-series experiments, image-based sampling and extrapolation, and real-world environmental data demonstrate its advantage over existing baselines, e.g., ConvNP and ANP.

Strengths: This is very nice work about Neural Process, which improves Neural Processes (NPs) with translation equivariance and extends convolutional conditional NPs to learn more expressive distribution. Both aspects are important to model more realistic probalistic distrition. Emprical evaluation is thorough and good analysis with experimental results. This is a significant contribution to this area. Overally speaking, this is a good paper.

Weaknesses: Almost none. It would be great if the authors can apply it on more challenging tasks and datasets, e.g., few shot learning. This will greatly strengthen the paper.

Correctness: It seems claims, method and the empirical methodology are correct.

Clarity: This paper is well written and well motivated. The author provides nice explanation for their proposed model and derivation, and the paper is easy to read and follow.

Relation to Prior Work: Yes, this paper provides good discussion with related works and connection with existing works.

Reproducibility: Yes

Additional Feedback: Rebuttal: I have read rebuttal and keep score


Review 4

Summary and Contributions: The paper aims to achieve improved modeling of stationary stochastic processes. The claimed contributions are the following: (1) Proposing a Convolutional Neural Process (ConvNPs) which extends Neural Processes (NPs) to use convolution, and extends Convolutional Conditional Neural Processes (ConvCNPs) by introducing latent variables. (2) Proposing a new biased, maximum likelihood objective for training NPs (to replace the ELBO which is typically used for training NPs). (3) Demonstrates improved performance on multiple tasks (e.g. 1D regression, image completion, a real world spatial data task).

Strengths: (1) The authors argue for combining the benefits of ConvCNP (translation equivariance) with those of NPs (more expressivity via latent variables) and provide a thorough discussion on the advantages. (2) The paper provides a good empirical investigation, comparing the proposed approach with the alternatives, including on a downstream task (albeit a toy problem), and also comparing the maximum likelihood vs. the ELBO objectives in isolation.

Weaknesses: (1) The approach itself is a combination of two proposed approaches (ConvCNP, NPs), so it might have limited impact from the methods perspective. That said, it's a good investigation into the architecture and training objective for NPs. (2) The approach considers baselines such as GP and ANP and ConvCNP. However, not all baselines are evaluated on all tasks. E.g. ANP and ConvCNP are not evaluated on the real data task, nor on the downstream task. Showing that ConvNP is consistently better than alternatives across tasks would improve the paper.

Correctness: The paper seems methodologically sound. The proposed approach is evaluated empirically on multiple tasks, with both illustrative and numerical results.

Clarity: The paper is well written; it describes clearly the approach, prior work and relevant terms. These are some minor comments: * In the notation section, the paper should describe what context and target sets are. These are explained later (e.g. in 2.1) but they should be described as soon as they are introduced. * I like the rainfall example. It would be nice if the example also illustrated the source of randomness in the random function from \mathcal{X} to \mathcal{Y}. * In Figure 5, the legend reads 'NP UCB' and 'BP TS' even though the approach used is ConvNP.

Relation to Prior Work: The authors combine the benefits of ConvCNP (translation equivariance) with those of NPs (more expressivity with latent variables). The paper adequately distinguishes themselves from these works. I’m not familiar with work that makes the same claims.

Reproducibility: Yes

Additional Feedback: The improvement in log likelihood for the ML vs. NP objective is much higher in the ConvNP paper vs. the ANP paper (e.g. in Table 2). Do you have any intuition on this?

[Author Response · NeurIPS 2020]

We thank the reviewers for their considered reviews, and are pleased there is unanimous agreement on accepting the
paper. In particular, reviewers found the paper provides significant contributions to both the modelling and training
aspects of Neural Processes, is well written, and includes extensive and convincing empirical investigation. Since
submission we have made improvements to the paper. We first address reviewer feedback, then describe these.

**Reviewer feedback.** We thank all reviewers for their helpful and constructive comments. We do not have space to
address them all here, but will do so in the paper. We appreciate R1's suggestion to move Fig 1 in the supplement
and Algs 3 and 4 to the main body, and will apply this to the revision. R1 mentions our image experiments lack
multimodality, but note that we provide a demonstration of multimodality in Figs 3c & 3d. Nevertheless, we find in
practice that the ANP tends to produce more diverse image samples: we will add a discussion of this point. R1 asks
why Eq 4 is not a valid ELBO. The correct ELBO is: $\mathbb{E}_{q(\mathbf{z}|D)}[\log p(\mathbf{y}|\mathbf{x}, \mathbf{z})] - \text{KL}(p(\mathbf{z}|D_c)\|q(\mathbf{z}|D))$. However, as
$p(\mathbf{z}|D_c)$ is intractable, this is *approximated* by $q(\mathbf{z}|D_c)$ (as described in Garnelo et al. [11] Eq. 9), hence this is no
longer a lower bound in a single consistent model. We will expand on this in the paper.

R3 asks if the method can be applied on more challenging tasks. The extension to, e.g. few-shot image classification is
interesting, but will require several modifications. As such, we leave this for future work. We agree with R4 that a
comparison with ANP and ConvCNP on the real-world experiment in Sec 5.3 would improve the paper. We did not
include it as our focus in this subsection was on translation equivariant models with coherent samples. It is unclear
how to apply the ANP effectively here, since the test regions do not overlap with the train region, and are of different
sizes. Further, the ConvCNP would not produce coherent samples, and cannot be applied to sampling in Fig 4 or to
Thompson sampling in Fig 5. R4 points out that the improvement in changing objective from $\mathcal{L}_{\text{NP}}$ to $\mathcal{L}_{\text{ML}}$ is more
significant for ConvNP than ANP. We believe this is because ConvNP has more latent variables than ANP (5000 vs
128), and, as alluded to on line 181, $\mathcal{L}_{\text{NP}}$ is more detrimental for models with more latent variables, since the KL-term
in Eq 5 is more of a 'distraction' from the max-likelihood target. We will expand on this in the paper.

**Heteroskedastic noise.** We have made a minor technical improvement which leads to significant performance gains:
changing the Gaussian observation noise from homoskedastic (hom. noise) $\sigma_y^2(\mathbf{z})$, to heteroskedastic (het. noise)
$\sigma_y^2(\mathbf{z}, \mathbf{x})$ in Eq 2. This change follows the findings of Le et al. [20], who demonstrate that het. noise improves
performance for several NP models, and is in line with Kim et al. [14]. We emphasise this is a design choice that does
not affect conceptual aspects of the paper or model. Moreover, we find this improves performance for both ConvNP and
ANP *and* is simpler to implement. Hence *we have rerun all experiments with het. noise and updated the paper.*

The results are: i) All models now perform better. An example with two kernels is provided in the table, comparing each
model with het. vs. hom. noise. The trends in the table generalise almost without exception in our experiments. ii) The
ordering of log-likelihood of ConvNP and ANP is unchanged in all experiments (including images): ConvNP ($\mathcal{L}_{\text{ML}}$)
still outperforms ANP. iii) As before, ANP fails catastrophically when spatially extrapolating. iv) ANP and ConvNP
both show tight predictions around training data. v) Other conclusions in the paper are unchanged. To summarise, this
change provides a strict improvement to the paper, as it simultaneously simplifies the implementation while leading to
significant improvements in performance in all settings considered in the empirical section.

In addition, we discovered a plotting error in Fig 2 in the submitted version, which as a result, depicted lower noise
variance for the ConvNP than should have been. We stress that this only affected *plotting* of the ConvNP in Fig 2, and
none of the surrounding results or analyses. Moreover, with the change to het. noise, both the ConvNP and ANP now
collapse their uncertainty around the data, and all of our results are replaced with improved versions. An example is
depicted in the figure below, where the top row details the corrected plots with hom. noise, and the bottom row shows
the models' predictives with the new het. noise on the Matérn-$\frac{5}{2}$ kernel. These plots should be compared with the 3rd
row of Fig 2 in the submission, which contains the plotting error for the ConvNP.

| | ConvNP (hom.) | ConvNP (het.) | ANP $\mathcal{L}_{\text{ML}}$ (hom.) | ANP $\mathcal{L}_{\text{ML}}$ (het.) | ANP $\mathcal{L}_{\text{NP}}$ (hom.) | ANP $\mathcal{L}_{\text{NP}}$ (het.) |
|---|---|---|---|---|---|---|
| Matérn-$\frac{5}{2}$ | $-0.80 \pm 7\text{E}{-}3$ | $\mathbf{-0.58} \pm 0.01$ | $-0.78 \pm 0.01$ | $\mathbf{-0.73} \pm 0.01$ | $\mathbf{-0.95} \pm 8\text{E}{-}3$ | $-0.96 \pm 0.01$ |
| Sawtooth | $1.22 \pm 0.01$ | $\mathbf{2.30} \pm 0.01$ | $-0.03 \pm 3\text{E}{-}3$ | $\mathbf{0.09} \pm 3\text{E}{-}3$ | $0.02 \pm 4\text{E}{-}3$ | $\mathbf{0.20} \pm 9\text{E}{-}3$ |



[Meta-Review · NeurIPS 2020]

Four knowledgeable referees support accept and I accept. We encourage and expect the authors to incorporate the reviewers' suggestions for improving the paper.